# Allelic effects on *KLHL17* expression underlie a pancreatic cancer genome-wide association signal at chr1p36.33

Katelyn E. Connelly [1], Katherine Hullin [1], Ehssan Abdolalizadeh[1], Jun Zhong[1], Daina Eiser[1], Aidan O'Brien [1], Irene Collins[1], Sudipto Das[2], Gerard Duncan[2], Pancreatic Cancer Cohort Consortium*, Pancreatic Cancer Case-Control Consortium*, Stephen J. Chanock [3], Rachael Z. Stolzenberg-Solomon[4], Alison P. Klein [5,6], Brian M. Wolpin [7], Jason W. Hoskins [1], Thorkell Andresson[2], Jill P. Smith [8] & Laufey T. Amundadottir [1]

Pancreatic Ductal Adenocarcinoma (PDAC) is the third leading cause of cancer-related deaths in the U.S. Both rare and common germline variants contribute to PDAC risk. Here, we fine-map and functionally characterize a common PDAC risk signal at chr1p36.33 (tagged by rs13303010) identified through a genome wide association study (GWAS). One of the fine-mapped SNPs, rs13303160 (OR = 1.23 (95% CI 1.15-1.32), *P*-value = 2.74×10$^{-9}$, LD r$^2$ = 0.93 with rs13303010 in 1000 G EUR samples) demonstrated allele-preferential gene regulatory activity in vitro and binding of JunB and JunD in vitro and in vivo. Expression Quantitative Trait Locus (eQTL) analysis identified *KLHL17* as a likely target gene underlying the signal. Proteomic analysis identified KLHL17 as a member of the Cullin-E3 ubiquitin ligase complex with vimentin and nestin as candidate substrates for degradation in PDAC-derived cells. In silico differential gene expression analysis of high and low *KLHL17* expressing GTEx pancreas samples suggested an association between lower *KLHL17* levels (risk associated) and pro-inflammatory pathways. We hypothesize that KLHL17 may mitigate cell injury and inflammation by recruiting nestin and vimentin for ubiquitination and degradation thereby influencing PDAC risk.

Pancreatic cancer is currently the third leading cause of cancer-related deaths and is expected to move to second place by 2030 in the United States[1]. Pancreatic ductal adenocarcinoma (PDAC) comprises over 90% of pancreatic cancer cases. While its survival rate has improved over the years, detection, prevention, and treatment of PDAC remains a challenge[2]. Epidemiological factors known to increase the risk of PDAC include Type 2 diabetes (T2D), pancreatitis, smoking, and obesity[3]. Additionally, both rare high-risk and common, low effect size germline variants are known to contribute to PDAC susceptibility[4–8].

[1]Laboratory of Translational Genomics, Division of Cancer Epidemiology and Genetics, National Cancer Institute, Rockville, MD, USA. [2]Protein Characterization Laboratory, Frederick National Laboratory for Cancer Research, Leidos Biomedical Research Inc, Frederick, MD, USA. [3]Laboratory of Genomic Susceptibility, Division of Cancer Epidemiology and Genetics, National Cancer Institute, Rockville, MD, USA. [4]Metabolic Epidemiology Branch, Division of Cancer Epidemiology and Genetics, National Cancer Institute, Rockville, MD, USA. [5]Department of Oncology, Sidney Kimmel Comprehensive Cancer Center, Johns Hopkins School of Medicine, Baltimore, MD, USA. [6]Department of Pathology, Sol Goldman Pancreatic Cancer Research Center, Johns Hopkins School of Medicine, Baltimore, MD, USA. [7]Department of Medical Oncology, Dana-Farber Cancer Institute, Boston, MA, USA. [8]Department of Medicine, Georgetown University, Washington, USA.*Lists of authors and their affiliations appear at the end of the paper. ✉e-mail: katelyn.connelly@nih.gov; amundadottirl@nih.gov

The Pancreatic Cancer Cohort Consortium and Pancreatic Cancer Case-Control Consortium have sought to identify common germline variants that influence risk of PDAC through genome-wide association studies (GWAS). Previous GWAS phases, PanScan I[5], II[6], and III[7,8] and PanC4[9], have identified 17 independent risk signals for PDAC. A 2018 meta-analysis of these four studies (9040 cases and 12,496 controls) and TaqMan replication using samples from the PANcreatic Disease ReseArch (PANDoRA) consortium (2,497 cases and 4611 controls) uncovered five new risk loci[4]. One of the newly identified loci was at chr1p36.33. The initial meta-analysis of PanScan I, II, III and PanC4 identified chr1p36.33 (tagged by single nucleotide polymorphism (SNP) rs13303010; $P$-value = 7.3 × 10$^{-7}$; odds ratio (OR) = 1.20; 95% confidence interval (CI) = 1.12–1.29) as a suggestive risk locus for PDAC (Table 1). A meta-analysis including samples from the PANDoRA consortium (11,537 cases and 17,107 controls) improved the signal beyond the GWAS significance threshold (rs13303010, OR = 1.26, 95% CI 1.19–1.35, $P$-value = 8.36 × 10$^{-14}$) (Table 1)[4], further supporting chr1p36.33 as a PDAC risk locus.

GWAS have been instrumental in estimating disease risk, identifying candidate genes, and uncovering novel pathways underlying disease development. However, functional characterization of GWAS loci is critical to pinpoint the biological mechanism underlying risk. Most GWAS signals map to non-coding, regulatory regions of the genome and are hypothesized to influence disease risk through allele-specific changes in gene expression[10]. Further, each locus is decorated with tens to hundreds of highly correlated variants complicating the identification of functional variant(s) and gene(s) underlying the risk signal. Statistical fine mapping and genomic assays (e.g., ATAC-seq and massively parallel reporter assays) have been beneficial for reducing the number of candidate functional variants to move forward for testing[11–13]. Chromatin capture and expression quantitative trait locus (eQTL) analysis are valuable for identifying putative functional genes[14,15]. While these methods greatly assist in the process of functionally characterizing GWAS risk loci, it is still a time-consuming process. In fact, the biological mechanisms underlying only a handful of the 22 published PDAC risk signals have been functionally characterized to date: chr5p15.33/*TERT*[16], chr16q23.1/*CTRB2*[17], and chr13q22.1/*DIS3*[18].

Interestingly, some overlap has been observed between GWAS loci for PDAC and associated epidemiological risk factors. A number of PDAC risk loci have common SNPs or colocalize with GWAS for traits that influence PDAC risk: chr1q32.1/*NR5A2*/T2D, chr2p13.3/*ETAA1*/T2D/waist-hip-ratio (obesity measure), chr8q24.21/*MIR1208*/T2D, chr9q34/*ABO*/T2D/body fat percentage, chr12q24.31/*HNF1A*/Maturity-onset Diabetes of the Young/T2D, chr16q23.1/*BCAR1*/T2D, chr18q21.32/*GRP*/T2D/BMI/obesity (https://mvp-ukbb.finngen.fi/ and NHGRI GWAS Catalog[19]). Further pathway enrichment analysis of genes ±100 kb of PDAC GWAS risk loci indicate an enrichment of genes associated with Maturity-onset diabetes of the young (KEGG), Sequence-specific DNA-binding transcription factor activity (GO Molecular Function), and Cellular response to UV (GO Biological Process)[4]. Further, DEPICT enrichment analysis indicated that genes associated with GWAS risk loci are highly expressed in numerous gastrointestinal tissues[4].

Here, we apply fine-mapping methods to identify candidate functional variants for in vitro testing. We subsequently identify rs13303160 as a functional variant at chr1p36.33 likely mediating the expression of *KLHL17* through allele-preferential binding of JunB and JunD transcription factors. Our work to characterize *KLHL17's* function in PDAC risk suggests a role in mitigating acinar to ductal metaplasia (ADM) and epithelial to mesenchymal transition (EMT) through protein homeostasis.

## Results

### Fine-mapping of the chr1p36.33 PDAC risk locus

To identify candidate functional variants at the chr1p36.33 risk locus, we first performed a meta-analysis using GWAS data from PanScan I-III[5–8], PanC4[9] and an additional 1066 PDAC cases and 9399 controls from the UK Biobank (UKBB)[20] resulting in 10,106 cases and 21,895 control subjects with imputed GWAS data. In this analysis, rs13303010 remained the most statistically significant SNP at chr1p36.33 (OR = 1.24, 95% CI 1.16–1.32, $P$-value = 2.09 × 10$^{-10}$) (Fig. 1, Table 1).

To identify credible causal variants (CCVs) underlying this association signal, we applied fine-mapping approaches to the summary statistics for this meta-analysis. We implemented the Bayesian approach Sum of Single Effects Linear Regression (SuSiE)[13] to identify credible sets of CCV with 90% confidence. SuSiE identified one credible set with five variants (Table 2). We also applied likelihood ratio (LLR < 1:100) and linkage disequilibrium (LD $r^2$ > 0.8) thresholds which identified two additional CCVs (Table 2). To be inclusive, we moved all seven variants forward for in vitro functional analysis.

Most GWAS variants are noncoding and thought to affect gene expression of target genes in an allele specific manner. As such, GWAS variants have been shown to be enriched in active gene regulatory elements indicated by posttranslational modifications of histones (H3K4me3, H3K4me1, H3K27ac) and accessible chromatin[11,21–23]. We examined the set of statistically fine-mapped variants in the context of maps of pancreas gene regulatory elements (using a ChromHMM 8-state model and ATAC-seq) we generated in PDAC and normal-derived pancreas cell lines[24]. All the fine-mapped variants at chr1p36.33 lie within active and bivalent transcriptional start sites and active enhancer elements (Fig. 1b, c). Two variants lie within regions of open chromatin (Fig. 1c, Table 2) lending further support for the fine-mapped variants as candidate functional SNPs influencing gene expression in *cis* or *trans*.

## Table 1 | Summary statistics for rs13303010 on 1p36.33 PDAC risk locus

| [a]SNP, [b]alleles, [c]chr, [d]location, and [e]gene | Study | MAF | | Subjects | | P-value | Allelic OR (95% CI) |
|---|---|---|---|---|---|---|---|
| | | Controls | Cases | Controls | Cases | | |
| rs13303010 (G, A) | 2018 Meta Analysis[4] | | | 12,496 | 9040 | 7.3 × 10$^{-7}$ | 1.20 (1.12–1.29) |
| 1p36.33 (894,573) | PANDoRA[4] | 0.10 | 0.14 | 4611 | 2497 | 6.0 × 10$^{-10}$ | 1.45 (1.33–1.57) |
| *NOC2L* | Meta Analysis + PANDoRA[4] | | | 17,107 | 11,537 | 8.36 × 10$^{-14}$ | 1.26 (1.19–1.35) |
| | UK Biobank | 0.09 | 0.12 | 9399 | 1066 | 1.05 × 10$^{-5}$ | 1.42 (1.21–1.65) |
| | Meta Analysis[4] + UK Biobank | | | 21,895 | 10,106 | 2.09 × 10$^{-10}$ | 1.24 (1.16–1.32) |

Summary statistics from previously published PDAC GWAS and for the new meta-analysis presented here including UK Biobank PDAC cases and controls for rs13303010 including minor allele frequencies in cases and controls, $P$-value, and odds ratio (OR) with the 95% confidence intervals (CI). Statistics are generated using a logistic regression model adjusted for covariates outlined in the Methods. $P$-values are not multiple-testing corrected. Source data are provided as a Source Data file.
[a]rsID; SNP ID number.
[b]Minor, major alleles.
[c]Location.
[d]Position (Genome build hg19).
[e]Nearest gene.

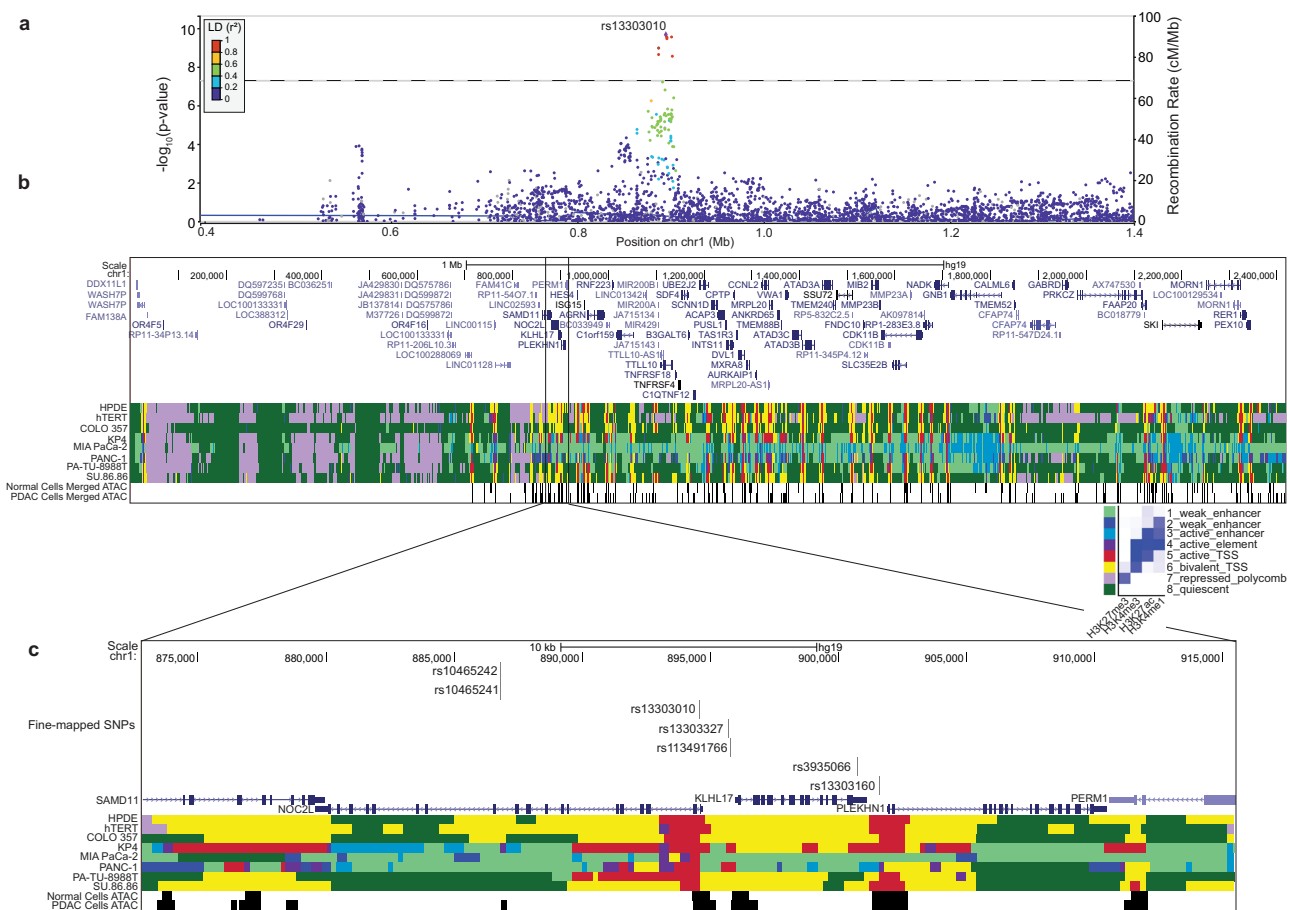

**Fig. 1 | Overview of chr1p36.33 PDAC risk locus. a** Locus Zoom plot of the variants identified in the meta-analysis, colors are indicative of the linkage disequilibrium (LD) $r^2$ in reference to the tag SNP (red $0.8 < r^2 \leq 1.0$; yellow $0.6 < r^2 < 0.8$, green $0.4 < r^2 < 0.6$, blue $0.2 < r^2 < 0.4$, purple $0 < r^2 < 0.2$), -log10 *P*-values are calculated using a logistic regression analysis and are not multiple-testing corrected. The gray-dashed line indicates the Bonferroni-corrected *P*-value significance threshold ($5 \times 10^{-8}$) used in GWAS; **b** UCSC genome browser view of chr1p36.33 with ChromHMM and ATAC-seq annotations in PDAC and normal-derived duct epithelial cell lines[24]; **c** Zoomed in UCSC browser snapshot showing the candidate functional variants and nearby genes. ChromHMM states are indicated by color: weak enhancer 1 (light green), weak enhancer 2 (dark blue), active enhancer (light blue), active element (purple), active transcriptional start site (TSS, red), bivalent TSS (yellow), polycomb repressed (light purple), quiescent (dark green). Source data are provided as a Source Data file.

## Table 2 | Summary statistics and fine-mapping for the CCVs in UK Biobank meta-analysis

| [a]SNP, [b]alleles, [c]location | OR (95% CI) | *P*-value | LD r2 | LLR | SuSiE PIP | Chromatin accessibility |
|---|---|---|---|---|---|---|
| rs13303010 (G, A) 1:894,573 | 1.24 (1.16-1.32) | $2.09 \times 10^{-10}$ | -- | 1.00 | 0.27 | Yes |
| rs3935066 (G, A) 1:900,730 | 1.25 (1.17-1.35) | $2.74 \times 10^{-10}$ | 0.89 | 1.30 | 0.22 | No (>500 bp) |
| rs113491766 (A, AG) 1:895,755 | 0.81 (0.76-0.86) | $2.85 \times 10^{-10}$ | 1 | 1.35 | 0.20 | ~56 bp away |
| rs13303327 (G, A) 1:895,706 | 1.24 (1.16-1.32) | $3.38 \times 10^{-10}$ | 1 | 1.60 | 0.18 | ~110 bp away |
| rs10465241 (C, T) 1:886,817 | 1.24 (1.16-1.32) | $1.03 \times 10^{-9}$ | 0.90 | 4.74 | 0.06 | ~22 bp |
| rs10465242 (G, A) 1:886,788 | 1.23 (1.15-1.32) | $2.23 \times 10^{-9}$ | 0.90 | 10.07 | 0.03 | ~52 bp |
| rs13303160 (G, A) 1:901,559 | 1.23 (1.15-1.32) | $2.74 \times 10^{-9}$ | 0.93 | 12.31 | 0.02 | Yes |

Summary statistics of the CCVs for the newest meta-analysis with PDAC UK Biobank cases and controls including OR and 95% CI, *P*-value, linkage disequilibrium (LD) $r^2$ reported relative to the tag SNP based on the European 1000Genomes reference, likelihood ratio (LLR), SuSiE posterior inclusion probability (PIP), and proximity to accessible chromatin. GWAS statistics are generated using a logistic regression model adjusted for covariates outlined in the Methods. *P*-values are not multiple-testing corrected. LLR was calculated using the GWAS *P*-values and a chi-squared test. Top five SNPs were identified in the SuSiE credible set of variants. All variants listed met the fine-mapping criteria for functional follow-up. Accessible chromatin is based on the ATAC-seq data generated in PDAC-derived or normal-derived pancreas epithelial cell lines[24]. Source data are provided as a Source Data file.
[a]rsID.
[b]Minor, major alleles.
[c]chr:position (Genome build hg19).

## Assessing allele-preferential binding and gene regulatory activity of candidate functional variants

To identify functional variants underlying this GWAS signal, we first sought to identify variants that exhibited allele-preferential transcription factor (TF) binding. We tested the set of seven fine-mapped variants at chr1p36.33 in electrophoretic mobility shift assays (EMSAs) using nuclear extracts from the PANC-1 pancreatic cancer cell line. Three of the seven variants demonstrated allele-preferential protein binding: rs13303010, rs13303327, and rs13303160 (Fig. 2). The GWAS tag SNP, rs13303010, showed preferential binding to the protective

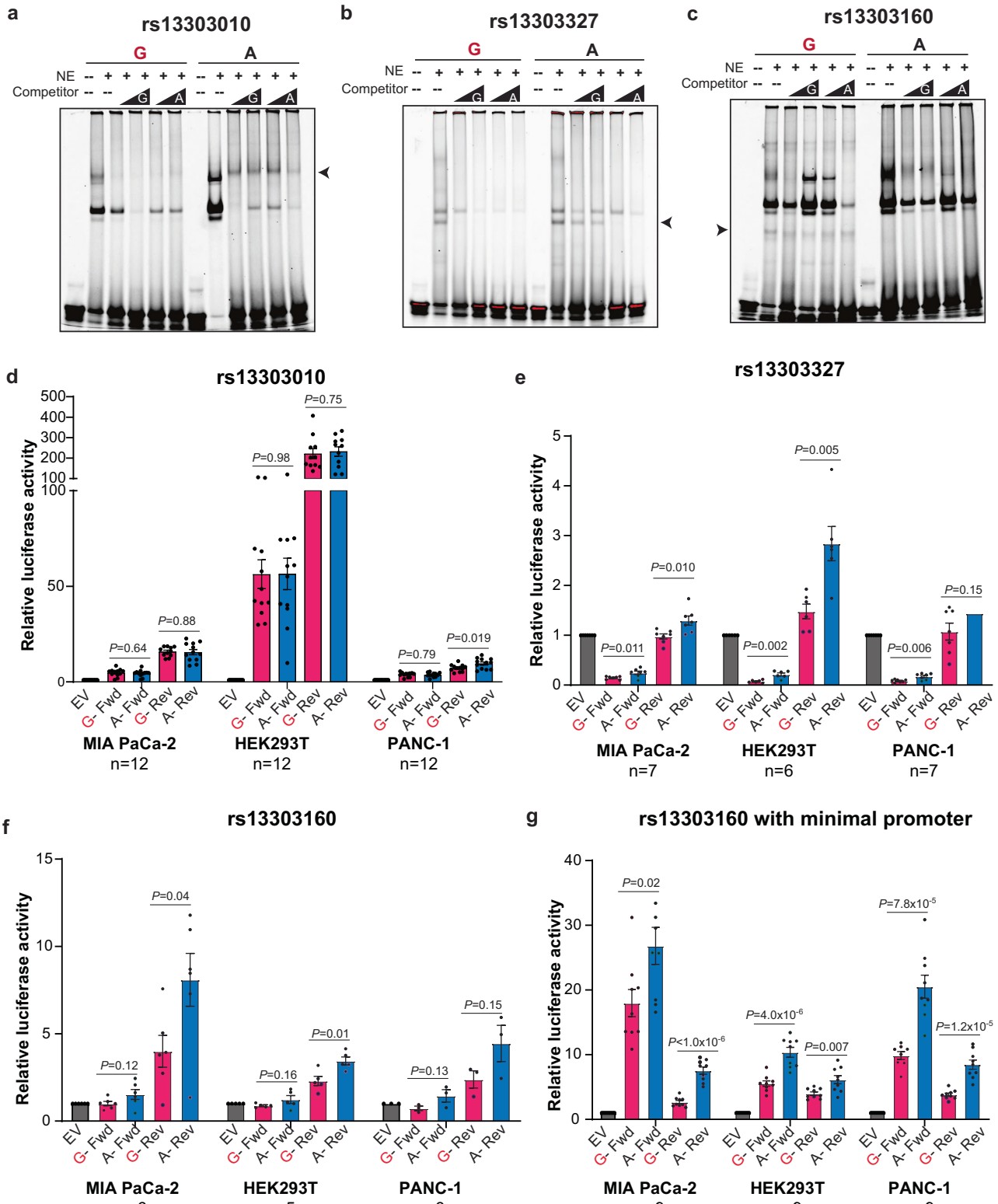

**Fig. 2 | Identification of allele-preferential binding and activity using EMSA and luciferase reporter assays. a–c** Representative EMSA results with PANC-1 nuclear extract and fluorescently labeled 31 bp oligonucleotides with each variant, rs13303010 ($n = 4$ independent experiments), rs13303327 ($n = 3$ independent experiments), rs13303160 ($n = 4$ independent experiments), respectively, centered in the middle. Competitor is the same sequence with no fluorescent label in excess (50, 100X, indicated by the black triangles). Arrows denote the allele-preferential binding. **d–f** Luciferase reporter assays using rs13303010, rs13303327, and rs13330160, respectively and the surrounding sequence as a promoter to the luciferase gene in three cell lines, number of biological replicates are indicated

below the cell line name. **g** Luciferase assay for rs13303160 and surrounding sequence as an enhancer upstream of a minimal promoter and luciferase gene in three cell lines. For EMSA and luciferase, the risk allele is colored in red. For luciferase, the forward (fwd) and reverse (rev) orientation of the sequence was used. Pink bars indicate the risk allele and blue bars the protective allele. Gray bars are the Empty Vector (EV) control. The number of biological replicates is indicated underneath the cell line in the figure for (**d–g**). Error bars represent the standard error of the mean (SEM) and significance was determined by an unpaired, two-tailed t-test. Source data are provided as a Source Data file.

**Table 3 | Predicted TFs with allelic binding preferences for rs13303327 and rs13303160**

| rsID | Motif | Allele 1 | Allele 2 | P-value Allele 1 | P-value Allele 2 | P-value fold change |
|------|-------|----------|----------|-------------------|-------------------|----------------------|
| rs13303327 | ELF1 | G | A | $2.27 \times 10^{-3}$ | $1.28 \times 10^{-4}$ | 17.74 |
| rs13303327 | ELF2 | G | A | $2.08 \times 10^{-3}$ | $8.70 \times 10^{-5}$ | 23.91 |
| rs13303327 | ELF3 | G | A | $1.81 \times 10^{-3}$ | $1.31 \times 10^{-4}$ | 13.83 |
| rs13303327 | ELF5 | G | A | $6.56 \times 10^{-4}$ | $2.66 \times 10^{-5}$ | 24.64 |
| rs13303160 | FOSB | G | A | $1.18 \times 10^{-4}$ | $2.93 \times 10^{-5}$ | 40.11 |
| rs13303160 | FOS | G | A | $5.73 \times 10^{-4}$ | $2.74 \times 10^{-5}$ | 20.86 |
| rs13303160 | FOSL1 | G | A | $4.32 \times 10^{-4}$ | $4.94 \times 10^{-6}$ | 87.31 |
| rs13303160 | FOSL2 | G | A | $8.16 \times 10^{-4}$ | $2.15 \times 10^{-5}$ | 37.93 |
| rs13303160 | JUNB | G | A | $8.00 \times 10^{-4}$ | $2.47 \times 10^{-5}$ | 32.36 |
| rs13303160 | JUND | G | A | $9.47 \times 10^{-4}$ | $2.28 \times 10^{-5}$ | 41.63 |
| rs13303160 | JUN | G | A | $8.40 \times 10^{-4}$ | $2.68 \times 10^{-5}$ | 31.37 |

Predicted TF motifs disrupted by SNP alleles for either rs13303327 or rs13303160. Alleles 1 and 2 are defined and P-values of predicted binding strength are indicated for each allele. P-values were calculated using the Perfectos-Ape algorithm[25]. The fold-change between the two P-values was calculated and used to determine the best predictions. Full prediction results are listed in Supplementary Table 1.

alternate allele (A) (Fig. 2a). Additionally, we observed consistent allele-preferential binding with the protective alternate allele (A) at rs13303327 (Fig. 2b). The third variant, rs13303160, exhibited some preferential binding to the risk-increasing reference (G) allele (Fig. 2c). These preferential binding patterns were also observed with MIA PaCa-2 (Supplementary Fig. 1a–c) and HeLa nuclear lysate (Supplementary Fig. 1d, e).

We next moved the three variants with allele-preferential binding forward to evaluate allele-preferential gene regulatory activity using luciferase reporter activity assays. As these variants lie near active transcriptional start sites (TSS) (Fig. 1c), we first tested allele-preferential promoter activity by placing 141-201 base pair (bp) sequences centered on the variant of interest (see Methods) into a luciferase vector without a minimal promoter (in the pGL4.14 vector). We observed strong promoter activity for the rs13303010 constructs compared to the empty vector (EV) but minimal allele-preferential luciferase activity in MIA PaCa-2, PANC-1 and HEK293T cells (Fig. 2d). The second SNP, rs13303327, demonstrated allele-preferential luciferase activity with the alternate (A) allele having a stronger regulatory effect (Fig. 2e). The third SNP, rs13303160, exhibited strong promoter activity with the alternate (A) allele demonstrating an allele-preferential effect in the reverse orientation in the MIA PaCa-2 and HEK293T cells (Fig. 2f).

While rs13303010 and rs13303327 lie in a 1328 bp region between the TSS of the *NOC2L* and *KLHL17* genes, rs13303160 is located 360 bp downstream of the 3′UTR of *KLHL17* and 303 bp upstream of the TSS for *PLEKHN1* (Fig. 1c), suggesting it could influence promoter and/or enhancer activity at this locus. We therefore tested the sequence surrounding rs13303160 as an enhancer upstream of a minimal promoter and the luciferase gene (in the pGL4.23 vector). As an enhancer element, the rs13303160 constructs demonstrated strong enhancer activity in all three cell lines with the alternate (A) allele exhibiting stronger activity (Fold change (FC) 1.5–2.8, P-value = $1 \times 10^{-6}$–0.02; Fig. 2g). Thus, through EMSA and luciferase assays, we narrowed the set of seven fine-mapped candidate functional variants down to two (rs13303327 and rs13303160) that each demonstrate allele-preferential protein binding and gene regulatory activity.

**Identifying allele-preferential protein binding**

To identify TFs potentially mediating the allele-preferential regulatory activity we observed, we performed an in silico TF motif analysis using PERFECTOS-APE[25]. This analysis predicts TF binding potential for both alleles of a variant and provides a P-value for the estimated strength of binding. We then calculated the fold-change between P-values as a proxy for the binding affinity change between alleles.

For rs13303327, we identified several E74-like factors (ELF), members of the E-twenty-six (ETS) family of transcription factors,

having a 13-24-fold difference in binding P-values between the A and G alleles (Table 3, Supplementary Table 1). The ELF transcription factors recognize the motif GGAA (Fig. 3a) which is disrupted by the risk allele-G at rs13303327 by replacing the last A with a G.

To validate TF binding predictions for rs13303327, we performed EMSAs with recombinant ELF1, ELF2, ELF3, and ELF4 proteins to assess allele-preferential binding in vitro (Fig. 3b). ELF2 consistently demonstrated preferential binding to the A allele as compared to the G allele (Fig. 3b, Supplementary Fig. 2). ELF1, 3 and 4 did not demonstrate consistent or differences in allelic binding (Fig.3b). Additionally, including an ELF2 antibody led to a loss of the observed binding further confirming ELF2 as the protein responsible for the allele-preferential band in the original EMSAs (Fig. 3c).

To assess if ELF2 is enriched at rs13303327 in an allele-preferential manner in the context of the native DNA, we performed chromatin immunoprecipitation followed by quantitative PCR (ChIP-qPCR) in two pancreatic cancer cell lines heterozygous at this SNP (Hs766T and SW1990). We were unable to enrich ELF2 at rs13303327 or a predicted positive control region from the SW1990 PDAC cell line. In Hs766T cells, we observed minimal enrichment of ELF2 relative to the IgG and negative control region (Fig. 3d). We then assessed allele-preferential enrichment of the ChIP DNA using a TaqMan genotype probe for rs13303327 and observed an enrichment of ELF2 over IgG; however, an allele-preferential enrichment compared to the input DNA was not noted (Fig. 3e). Based on these results, we conclude that ELF2 does not bind rs13303327 in an allele-preferential manner in PDAC cell lines.

For rs13303160, allele-preferential binding prediction pointed towards preferential binding of TFs to the A allele, which was opposite of what we observed in the EMSA (Fig. 2c) but consistent with the luciferase assay (Fig. 2f, g). The transcription factors with the largest fold change in predicted binding P-values (20-87-fold) were the Activator Protein 1 (AP-1) transcription factors (Jun and Fos) (Table 3 and Supplementary Table 1) with the risk allele (G) disrupting the AP-1 motif (Fig. 4a).

The binding motif for AP-1 (and sequence flanking rs13303160) is a TPA (12-O-Tetradecanoylphorbol-13-Acetate) response element indicating that TPA may induce expression and binding of AP-1 to these elements[26]. We therefore repeated both the EMSA and luciferase experiments using cells treated with TPA. Upon TPA treatment, we observed an induction of AP-1 as demonstrated by the increase in JunB protein expression (Supplementary Fig. 3). EMSAs with TPA-treated HeLa nuclear extract demonstrated allele-preferential binding opposite what was originally seen (Fig. 2c) with preferential binding to the A allele over the G allele (Fig. 4b). In luciferase assays, cells treated with TPA demonstrated a stronger induction of enhancer activity in the

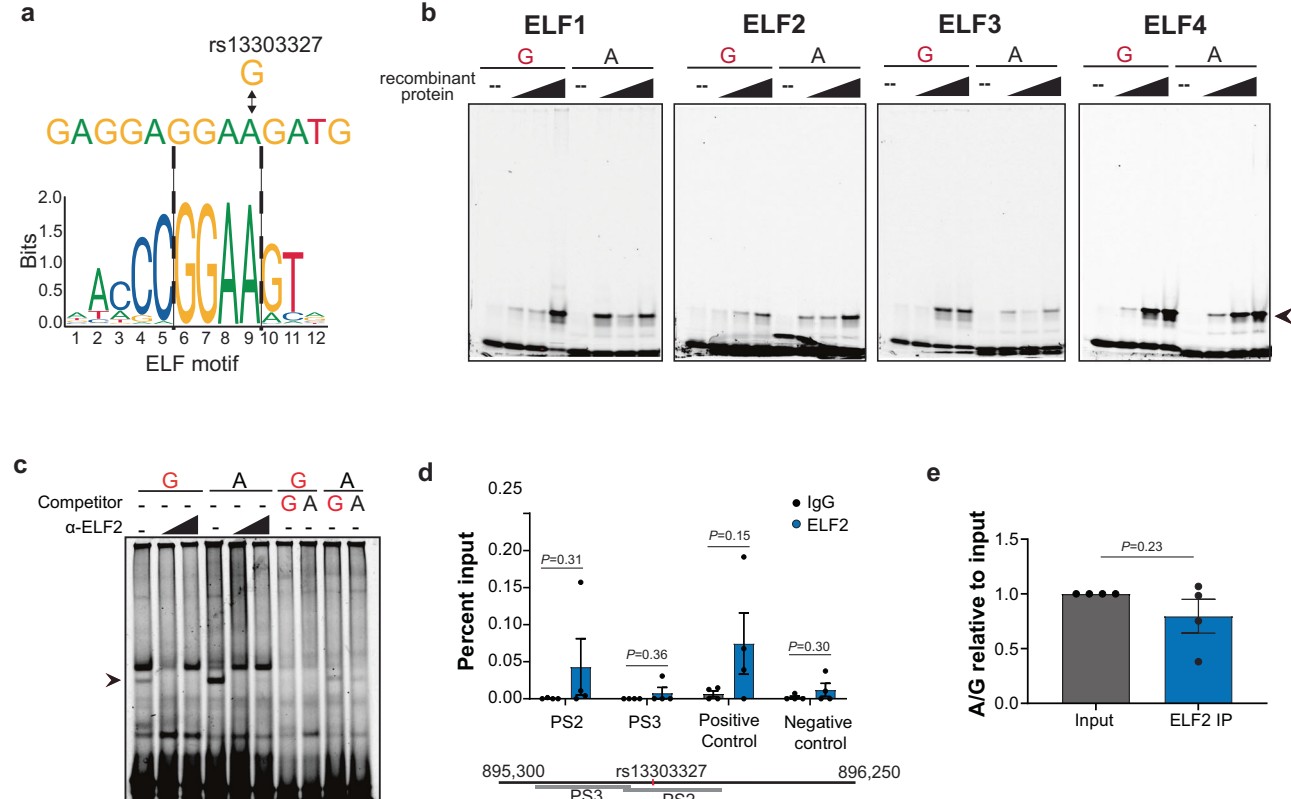

**Fig. 3 | Allele-preferential binding of ELF to rs13303327 in vitro and in vivo. a** in silico transcription factor (TF) binding motif prediction for rs13303327. The risk G allele disrupts the motif. **b** Representative EMSA images with increasing amounts of recombinant ELF proteins (indicated by black triangles) and fluorescently labeled oligonucleotide (n = 4 (ELF1), 4 (ELF2), 3 (ELF3), 3 (ELF4) individual experiments); arrow denotes binding of interest. The risk allele is denoted in red. **c** Representative EMSA supershift with an antibody against ELF2 using PANC-1 nuclear lysate (n = 3 biological replicates). The risk allele is indicated in red. **d** ChIP-qPCR for ELF2 in Hs766T PDAC cell line with primers near or encompassing the SNP. The positive control is a documented region from an ELF2 ChIP-seq in K562 (ENCODE) and negative control is from a quiescent region of chr1p36.33. Percent input enrichment was quantitatively determined using a standard curve derived from input DNA as described in the ActiveMotif protocol. Gray bars indicate IgG, blue bars indicate ELF2, n = 4 biological replicates. **e** TaqMan genotyping of enriched ChIP-qPCR DNA; A to G ratio was calculated relative to the input ratio. Gray bar is input and blue bar is ELF2 IP, n = 4 biological replicates. Error bars represent SEM. All statistical tests were unpaired, two-tailed t tests were performed. Source data are provided as a Source Data file.

PANC-1 cell line compared to vehicle control (Fig. 4c). However, the allele-preferential effects remained the same with the protective A allele showing higher activity as compared to the risk increasing G allele (FC 2-3.9, P-value = $6.3 \times 10^{-3}$–0.018). This indicates that AP-1 proteins may be responsible for the gene regulatory activity observed in the luciferase assays.

The AP-1 family of proteins includes the c-Fos and c-Jun proteins that can homo- or heterodimerize and play different roles in transcriptional regulation depending on the context[26]. The in silico analysis did not predict which of the AP-1 protein family members bind rs13303160 as they all recognize the same motif. To determine which AP-1 protein(s) may exhibit allele-preferential binding in vitro, we performed EMSA using recombinant proteins for c-Fos, FosB, Fos-related antigen 1 (FRA-1), Fos-related antigen 2 (FRA-2) (Fig. 4d), Jun, JunB, and JunD (Fig. 4e). c-Fos and FosB demonstrated some allele-preferential binding, though inconsistently. Recombinant JunB and JunD proteins, on the other hand, demonstrated consistent allele-preferential binding. We subsequently performed supershift EMSAs with antibodies against JunB and JunD and observed a shift in the allele-preferential band when the antibody is added to the binding reaction indicating that the allele-preferential bands observed include JunB and JunD (Fig. 4f,g).

We then assessed if the in vitro allele-preferential binding of JunB and JunD translated to the context of genomic DNA. We performed ChIP-qPCR for JunB and JunD in the PDAC SW1990 and Hs766T cell lines (both heterozygous at rs13303160) and observed an enrichment of JunB with two primer sets that encompass the SNP and a third primer

set just upstream of the SNP in SW1990 cells (Fig. 4h). We were unable to observe consistent enrichment in the Hs766T cell line with these primers. We additionally examined JunD localization at rs13303160 in the SW1990 cell line and observed a significant enrichment of JunD with primer set 2 (PS2) that encompasses the SNP (Fig. 4h). To assess allele-preferential enrichment in SW1990 cells, we quantified the immunoprecipitated DNA using a TaqMan genotyping assay. Compared to the input DNA, we observed an increased enrichment of the A allele over the G allele for both JunB (FC = 5.2; P-value = $2.4 \times 10^{-3}$) and JunD (FC = 3.4 P-value = $1 \times 10^{-5}$) (Fig. 4i).

In summary, functional characterization of seven fine-mapped SNPs at chr1p36.33 led to the identification of three variants with allele-preferential binding in vitro, two of which also displayed allele-preferential gene regulatory activity. Transcription factor binding predictions and in vitro binding assays identified ELF2 and JunB/JunD as likely mediators of the allele-preferential effect at rs13303327 and rs13303160, respectively. Further in vivo analysis using ChIP-qPCR highlighted allele-preferential binding of JunB/D at rs13303160. Thus, we conclude rs13303160 represents a functional variant at the chr1p36.33 pancreatic cancer risk locus.

**Identification of likely target genes mediating risk at chr1p36.33**
Chr1p36.33 is gene dense with over 30 protein coding genes, micro-RNAs, and long non-coding RNAs. Using eQTL analysis, we identified two potential target genes at this locus for the tag SNP rs13303010 using the pancreas-specific GTEx (v8, n = 305) dataset: *KLHL17*

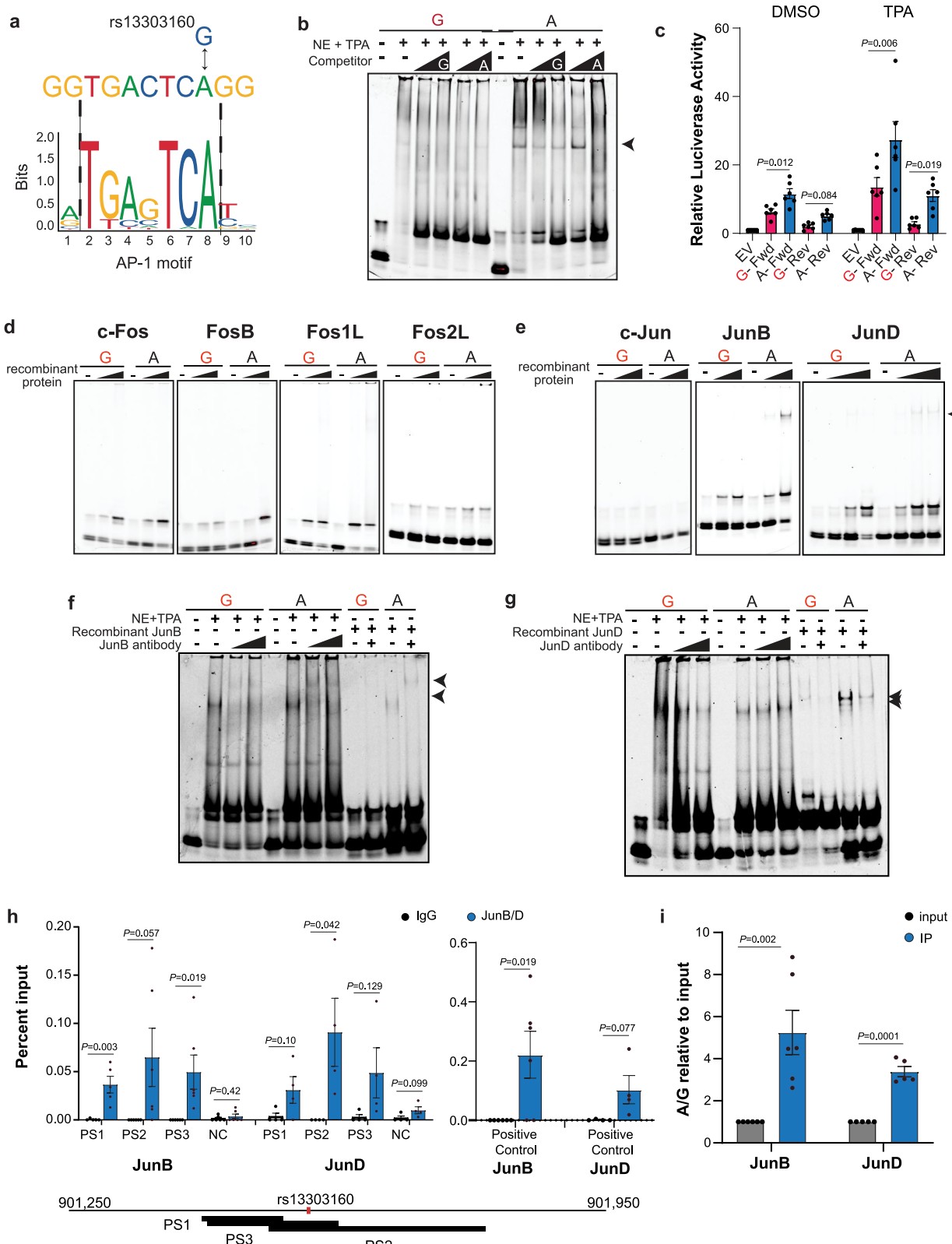

(Normalized Effect Size, NES = 0.35; P-value = 4.9 × 10⁻⁹) and *NOC2L* (NES = −0.25; P-value = 3.2 × 10⁻⁵) (Fig. 5a). Colocalization analysis[27] of the GTEx (v7) pancreas eQTL and the PDAC GWAS signals indicated that the *KLHL17* and *NOC2L* eQTL may share a single causal variant with the GWAS signal (posterior probability = 0.99 and 0.75, respectively). Further, a transcriptome-wide association study (TWAS) from our group identified *KLHL17* as a borderline significant gene in the pancreas and multi-tissue models (Z = −3.96, FDR = 0.052; FDR = 0.063, respectively)[28]. Due to the stronger colocalization probability, suggestive TWAS results, and corroborating in vitro allele-preferential activity, we focused on *KLHL17* as a likely target gene underlying the chr1p36.33 risk signal.

**Fig. 4 | Allele-preferential binding of AP-1 proteins to rs13303160 in vitro and in vivo. a** in silico Transcription factor binding predictions; the risk (G) allele disrupts the motif. **b** Representative EMSA using TPA-stimulated nuclear HeLa extract and fluorescently labeled oligonucleotide (*n* = 3 independent experiments). Arrow indicates the allele-preferential binding. **c** Luciferase reporter assay using DMSO and TPA stimulation and the rs13303160 sequence as an enhancer in the PANC-1 cell line; luciferase activity reported relative to the Empty Vector (EV, gray bars). Pink bars denote the risk allele, and blue bars represent the protective allele. Both alleles were tested in the forward (fwd) and reverse (rev) orientations. Unpaired, two-tailed *t* tests were performed on the relative luciferase activity of the A/G ratio compared to A/A; *n* = 6 biological replicates. **d** Representative EMSAs with increasing amounts of recombinant Fos proteins (from left to right: c-Fos (*n* = 3 independent experiments); FosB (*n* = 2); Fos1L (*n* = 1); Fos2L (*n* = 1)). **e** Representative EMSAs with increasing amounts of recombinant Jun proteins (from left to right: c-Jun (*n* = 2 independent experiments), JunB (*n* = 2), JunD (*n* = 2)).

Arrow indicates the allele-specific binding. **f, g** Representative supershift EMSA with antibodies against JunB and JunD (*n* = 3 independent experiments each), respectively, using both TPA-stimulated nuclear lysate and recombinant protein; Arrows denote the shift in the bands. **h** ChIP-qPCR in SW1990 PDAC cells for JunB (denoted in blue, IgG in gray) (*n* = 6 biological replicates) and JunD (denoted in blue, IgG in gray) (*n* = 4 biological replicates) using 3 primer sets (PS) surrounding the SNP. Positive control (PC) is from a JunB ChIP-seq performed in the CFPAC1 PDAC cell line[68]. Negative control (NC) is from a quiescent region on chr1p36.33; **i** TaqMan genotyping assay for rs13303160 using immunoprecipitated DNA from the ChIP. The ratio of A to G was determined relative to the quantity of A and G alleles in the input DNA (gray) (*n* = 6 biological replicates for JunB; *n* = 5 biological replicates for JunD, blue bars). Red font denotes the risk allele in (**b–g**). For all graphs, error bars represent the SEM. Unpaired two-tailed *t* tests were performed. Source data are provided as a Source Data file.

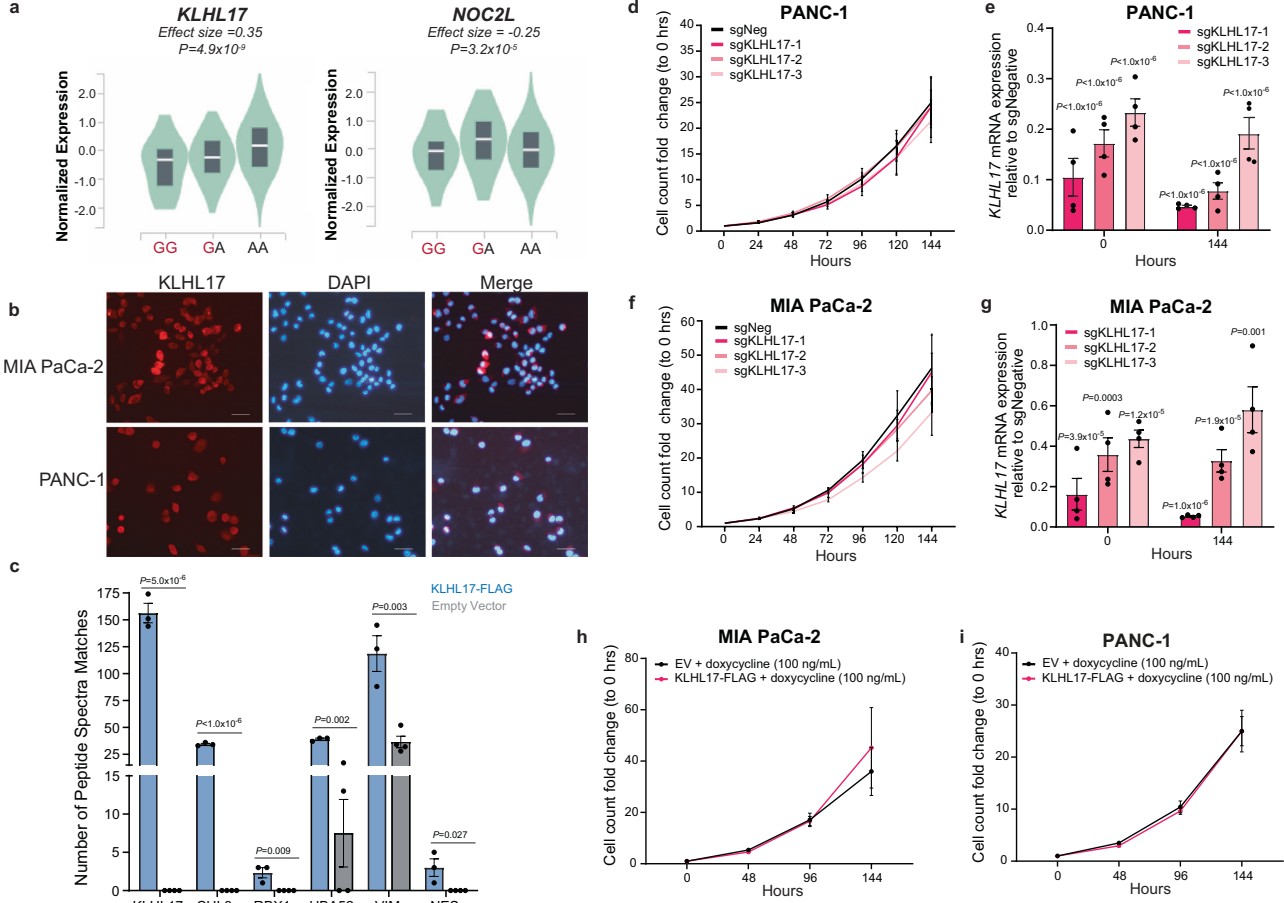

**Fig. 5 | Analysis of the effects of altered KLHL17, a Cullin3-E3 complex member, expression on cellular growth of PDAC cells in vitro. a** Pancreas GTEx v8 eQTLs for rs13303010, *n* = 20 (GG), 67 (GA), 218 (AA) individuals from the GTEx portal. Red denotes the risk allele. Linear regression using FastQTL was performed to identify the eQTLs and calculate the *P*-values (details on the GTEx webpage). The violin plots display the density distribution of the data with the white line indicating the median normalized expression, and the gray box displaying the interquartile range. **b** Representative immunofluorescence for KLHL17 in the MIA PaCa-2 (*n* = 2) and PANC-1 (*n* = 2) KLHL17 overexpressing cell lines. Scale bar = 100 micron. **c** Peptide Spectra Matches for Cullin3-E3 members identified by KLHL17-FLAG immunoprecipitation (*n* = 3 independent KLHL17-FLAG IPs, *n* = 4 independent Empty Vector IPs, ran multiplexed across 2 mass spectrometry runs) and mass spectrometry analysis. **d, f** Cell counts normalized to 0 h for CRISPRi-mediated knockdown of

*KLHL17* in PANC-1 (*n* = 4 biological replicates) and MIA PaCa-2 (*n* = 4 biological replicates) cells, respectively (left panel). sgNeg (black line) is the negative control targeting a sequence within the same topologically associated domain (TAD). Pink line represent three different sgRNAs; **e,g** qPCR analysis of CRISPRi sgRNA efficiency; expression is relative to the sgNeg control and internal *HPRT* control (right panel) and measured at the start and end of the growth assay (*n* = 4 biological replicates). **h, i** Cell counts normalized to 0 h for doxycycline-inducible KLHL17-FLAG overexpressing (pink) and empty vector (EV, gray) control MIA PaCa-2 (*n* = 4 biological replicates) and PANC-1 (*n* = 4 biological replicates) cell lines, respectively. For panels (**c–g**), error bars indicate the SEM; For panels (**c–g**), unpaired, two-tailed *t* tests were performed relative to the empty vector IP (**c**) or sgNeg (**d–g**). Source data are provided as a Source Data file.

## Characterizing the function of KLHL17 in the pancreas

KLHL17 is a member of the Kelch family of proteins. These proteins are substrate recognition proteins for the Cullin3-E3 ubiquitin complex (CRL3)[29], but the function of KLHL17 in the pancreas has not been defined. We therefore generated doxycycline-inducible KLHL17-FLAG tagged overexpressing PANC-1 and MIA PaCa-2 cell lines (Supplementary Fig. 4a, b) to study its function in the pancreas. We first examined KLHL17's cellular localization using immunofluorescence. In the MIA PaCa-2 and PANC-1 overexpressing cell lines, KLHL17 localized throughout the cell (Fig. 5b). This is in contrast to findings from the Human Protein Atlas in A-431 (epidermoid carcinoma), U-251MG (glioblastoma), and U2OS (osteosarcoma) cells, which indicate localization in the nucleoplasm and nuclear bodies[30] suggesting possible cell-type and/or context-specific functions for KLHL17. We then assessed whether KLHL17 is a member of the Cullin3-E3 ubiquitin complex. Using whole cell lysates from the PANC-1 induced overexpression and empty vector control cell lines, we performed an immunoprecipitation (IP) with a FLAG antibody (Supplementary Fig. 4c, d) followed by mass spectrometry (IP-MS) to identify proteins interacting with KLHL17. We identified enrichment of KLHL17 along with Cullin-3 (CUL3), E3 ubiquitin-protein ligase RBX1 (RBX1), and ubiquitin-ribosomal protein eL40 fusion protein (UBA52) in the KLHL17-FLAG IP but not in the empty vector control IP (Fig. 5c) indicating that KLHL17 is a part of the CRL3 ubiquitin ligase complex in pancreatic cells.

To identify candidate substrates for degradation by KLHL17, we applied a 1.5-fold enrichment filter (KLHL17-FLAG/Empty Vector) to the KLHL17 IP-MS data and identified 62 proteins (Supplementary Data 1). To further narrow this set of proteins down, we performed a global proteomics experiment where we titrated KLHL17 expression in our PANC-1 inducible overexpression system to examine the candidate protein(s) expression as KLHL17 expression levels increase. As KLHL17 is associated with the CRL3 ubiquitin complex, we would expect that KLHL17 substrates would decrease with increasing KLHL17 protein levels. This narrowed the list of candidate substrates to 23 proteins. Of interest are vimentin/VIM ($P$-value = 0.0029) and nestin ($P$-value = 0.027) (Fig. 5c, Supplementary Fig. 4e, f, Supplementary Data 1) as both have roles in early pancreatic carcinogenesis and are upregulated in PDAC[31–35]. In a follow-up global proteomics experiment using the MIA PaCa-2 KLHL17 overexpression system, we observed a downward trend in VIM with increasing KLHL17 expression, albeit not significant ($P$-value = 0.17) (Supplementary Fig. 4g, Supplementary Data 1). Nestin, on the other hand, did not display a decrease in expression with increasing KLHL17 (Supplementary Fig. 4h, Supplementary Data 1) in MIA PaCa-2 cells, possibly due to the fact that NES/nestin mRNA[30] and protein expression (Supplementary Fig. 4h) is 20-70 fold lower as compared to PANC-1 cells. This suggests that KLHL17 may recruit vimentin and nestin to the CRL3 ubiquitin ligase complex for ubiquitination and degradation.

## Assessment of cell growth after KLHL17 over-expression and knockdown

We next assessed if KLHL17 plays a role in pancreas cell proliferation and cell viability. Knockdown of KLHL17 using an siRNA pool in the PANC-1 and MIA PaCa-2 PDAC cell lines revealed a significant reduction in proliferation for both cell lines four days following knockdown (Supplementary Fig. 5a, c). However, we observed limited knockdown efficiency for KLHL17 and a non-specific reduction of the nearby NOC2L gene (Supplementary Fig. 5b, d). As these results did not distinguish whether the reduced levels of KLHL17 or NOC2L mediated the growth effects observed, we implemented CRISPR interference (CRISRPi) in PANC-1 and MIA PaCa-2 cells using guide RNAs targeting the 5' untranslated region (UTR) of KLHL17. CRISPRi-mediated knockdown of KLHL17 significantly reduced KLHL17 mRNA levels in PANC-1 and MIA PaCa-2 cell lines (Fig. 5e, g) while minimally affecting NOC2L expression (Supplementary Fig. 5e, f). Surprisingly, despite the significant reduction in KLHL17 mRNA levels, KLHL17 protein levels remain unchanged

even 36 days after transduction (Supplementary Fig. 5g, h). Subsequent growth analysis revealed that inhibition of KLHL17 mRNA expression did not affect cell proliferation in the two PDAC cell lines (Fig. 5d, f). Furthermore, overexpression of KLHL17 in PANC-1 and MIA PaCa-2 cells did not alter cell growth (Fig. 5h, i, Supplementary Fig. 4a). We hypothesize that the initial growth phenotype observed with KLHL17 siRNA is likely a result of the non-specific reduction in NOC2L expression as targeted knockdown of NOC2L with siRNA also reduces cell growth (Supplementary Fig. 5i–l).

## Interrogating the functional consequences of decreased KLHL17

We sought to investigate additional potential functional consequences of altered KLHL17 expression in the pancreas using an in silico knockdown analysis as previously described[28,36]. We performed this analysis using the GTExv8 Pancreas RNA-seq dataset. Samples were separated into quartiles based on KLHL17 gene expression and the highest ($n = 82$) and lowest ($n = 82$) quartiles (Supplementary Fig. 6a) were subjected to differential gene expression analysis with EdgeR[37]. We then used Gene Set Enrichment Analysis (GSEA)[38,39] and QIAGEN Ingenuity Pathway Analysis (IPA)[40] on the significantly differentially expressed genes (GSEA: $n = 13,090$, FDR < 0.05; IPA: $n = 3511$, FDR < 0.05 and log2 fold change > |0.5| (FC = 1.4)) to identify enriched pathways. We observed a positive enrichment of gene sets involved in inflammation or inflammation-related diseases in the group with the lowest KLHL17 expression (Fig. 6a–c). Further, IPA indicated similar results with nine of the ten most significant pathways being related to inflammation (Supplementary Fig. 6b) and eight of the nine pathways predicted to be activated in the GTEx samples with lower KLHL17 expression. This indicates that lower KLHL17 expression may be associated with a pro-inflammatory environment in the pancreas. We hypothesize that KLHL17 plays a role in mitigating cell injury and inflammation by recruiting vimentin and nestin for ubiquitination and degradation. This suggests that lower expression of KLHL17 that is associated with the risk promotes a pro-inflammatory environment that is prime for tumorigenesis (Fig. 6d).

## Discussion

Here, we functionally characterize a common PDAC risk locus at chr1p36.33. Fine mapping identified a set of seven candidate functional variants for further analysis. In vitro binding assays narrowed this set to three SNPs, namely rs13303010, rs13303327, and rs13303160 that exhibited allele-preferential protein binding. Subsequent luciferase experiments revealed in vitro allele-preferential gene regulatory activity for rs13303327 and rs13303160, but not rs13303010. We next identified ELF2 and JunB/D as transcription factors mediating the binding preference to the protective alleles of rs13303327 and rs13303160, respectively. This binding preference for JunB and JunD at rs13303160 was confirmed in the context of native chromatin. Taken together, we conclude that rs13303160 is a functional SNP at chr1p36.33 whose effect is mediated through allele-preferential binding of JunB and JunD.

eQTL analysis in GTEx pancreas samples identified two possible target genes, KLHL17 and NOC2L. NOC2L encodes the protein Novel INHAT Repressor which is known to repress p53 and histone acetyltransferase activity[41]. KLHL17 encodes Actinfilin, a known substrate recognition protein for the Cullin3-RING ligases, in neurons[42]. We focused on characterizing KLHL17 as a functional gene underlying the chr1p36.33 GWAS signal for multiple reasons: (1) strong colocalization of the KLHL17 eQTL with the GWAS signal; (2) KLHL17 is a suggestive PDAC TWAS gene[28]; and (3) allele-preferential luciferase regulatory activity and TF binding that were congruent with the KLHL17 eQTL.

The KLHL family, comprised of 42 proteins, has been reported to have countless roles in cancer including gastrointestinal cancers[29,43]. Canonically, KLHL proteins are substrate adaptor proteins for CRL3, an E3 ubiquitin ligase complex[44], that are responsible for mediating the

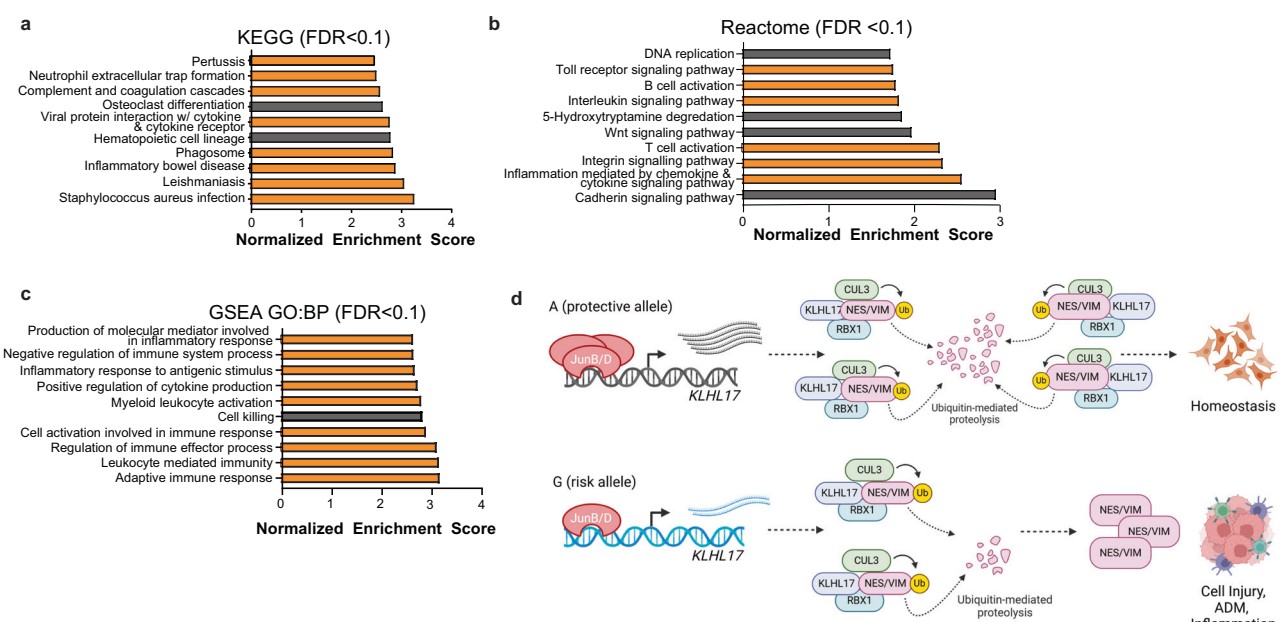

**Fig. 6 | Gene Set Enrichment Analysis (GSEA) of differentially expressed genes from an in silico *KLHL17* knockdown. a** GSEA using the KEGG functional database and the significantly (FDR < 0.05) differentially expressed genes when *KLHL17* expression is lower. **b** GSEA using the Reactome dataset. **c** GSEA using Biological Processes Gene Ontology dataset. For all three analyses only gene sets with a False Discovery Rate (FDR) < 0.1 are shown. Bars in orange indicate gene sets associated with inflammation or inflammation-related diseases. No gene sets were negatively enriched at this FDR threshold. Source data are provided as a Source Data file; **d** Working hypothesis for the function of rs13303160 and KLHL17 in PDAC risk. Created in BioRender. Connelly, K. (2025) https://BioRender.com/t14y333.

recognition, ubiquitination and degradation of their protein substrate(s) and are involved in a variety of cellular processes[29]. Changes in KLHL protein expression affect substrate protein expression and downstream pathways[45,46]. Depending on the KLHL family member and the cancer context, both increased and decreased KLHL family member expression has been described in cancer[29,43]. Recently, there has been an interest in targeting KLHL proteins to uncover their mechanisms and as a therapeutic strategy for associated diseases such as cancer[47].

Only one study has characterized KLHL17's role in cancer[48]. In Liu et al., the authors highlighted a non-canonical role for *KLHL17* in the Ras/Map Kinase (MAPK) pathway where *KLHL17* overexpression enhances cell proliferation, migration, and colony forming ability for non-small cell lung cancer (NSCLC)[48]. They did not indicate if the involvement with MAPK could be through protein ubiquitination, as is known for the KLHL proteins and KLHL17 in neurons[42]. In PDAC-derived cells, we observed that KLHL17 associates with members of the CRL3 family supporting a canonical function for this protein in the pancreas, as a CRL3 substrate recognition protein. We further identified candidate substrates that KLHL17 may recruit for ubiquitination and degradation. Two potential protein substrates of interest were the intermediate filament proteins vimentin and nestin, which have documented roles in inflammation[31,33,34], tumorigenesis[31,49], and metastasis[32,50–52].

Further, Liu and colleagues reported an overexpression of *KLHL17* in NSCLC tumors compared to adjacent-normal tissue from TCGA[48]. In the pancreas, there was no significant difference in *KLHL17* expression between TCGA PAAD tumor samples and normal or normal-adjacent tissue[35]. However, eQTL and TWAS results indicate that lower *KLHL17* expression levels are associated with increased PDAC risk[14,28]. We sought to characterize KLHL17's function on pancreas cell growth but were presented with several technical challenges: (1) our initial knockdown experiments using siRNA against *KLHL17* revealed a growth suppression, but knockdown efficiency was minimal and non-specific; (2) CRISPRi-mediated knockdown of *KLHL17*, which did not demonstrate a growth phenotype, reduced *KLHL17* mRNA expression,

however, protein levels remained unchanged, even after 36 days. This confounding finding means that we cannot rule out a growth phenotype for different KLHL17 protein levels. We hypothesize that the initial growth phenotype observed with *KLHL17* siRNA can be contributed to the non-specific knockdown of *NOC2L*, as specific knockdown of *NOC2L* using siRNA also suppressed cell growth. This suggests that *NOC2L* may play a role in PDAC tumorigenesis.

To uncover *KLHL17's* role as a functional gene underlying the chr1p36.33 risk signal, we utilized an agnostic approach to start to characterize its function in the pancreas. In silico differential gene expression analysis using GTEx pancreas samples comparing those with low *KLHL17* to high *KLHL17* mRNA expression highlighted an enrichment of upregulated genes involved in inflammation-related pathways and gene sets. This suggests that lower levels of *KLHL17* may associate with a pro-inflammatory state in the pancreas, or reduced ability to resolve the consequences of inflammatory signals. While such an in silico approach has been shown to have a strong concordance with knockout mouse RNA-sequencing data[36], there are limitations to consider when assessing the results. First, for *KLHL17*, there is a relatively small difference in expression between samples in the upper and lower quartile (~2 fold). Second, because the dataset is derived from 168 bulk pancreas tissue samples, heterogeneity (e.g. cell type composition, genetics, and environmental factors) is likely to add noise. However, this approach provided us with a basis to develop hypotheses regarding KLHL17's function in the pancreas.

Inflammation is a contributing risk factor to the development of PDAC[53]. PDAC arises from pancreatic intraepithelial neoplasia (PanIN) that display cancerous and pancreatic duct cell properties. Although PanIN exhibit duct-like properties, multiple lines of evidence indicate that pancreatic acinar cells that have undergone acinar-to-ductal metaplasia (ADM) are precursors for PDAC[54]. ADM is a trans-differentiation process in which acinar cells lose acinar-specific markers and gain duct cell markers. The plasticity of acinar cells makes them highly sensitive to external stimuli[55]. Acinar cells can recover from an acute stimulus, but with a more sustained stimulus, such as during chronic inflammation, ADM can become irreversible resulting

in progression to PanIN[56]. In addition to ADM, epithelial to mesench-ymal transition (EMT) is observed in pre-malignant PanIN lesions and is prevalent in regions of ADM with inflammation[31,57].

Vimentin and nestin expression, identified in our proteomics as candidate KLHL17 substrates, are induced upon cell injury and stress. Vimentin plays an important role in EMT, the recruitment of inflam-matory cells to resident tissues, the activation of the inflammasome, and fibrosis[58]. Nestin is a binding partner of vimentin and a marker of multi-potent progenitors. Upon stress and injury, nestin expression is induced and nestin-positive cells have the ability to re-enter the cell cycle and differentiate in the repair process[33,34,50,59]. In the pancreas, both proteins have been implicated in ADM and pre-malignant EMT and are upregulated in tumors[28,30,35]. During ADM following pancreatic injury, a transition population of nestin-positive cells is formed[34]. Additionally, changes in nestin expression correlate with changes in EMT markers[32,52]. Finally, induction of pancreatitis promotes EMT and vimentin expression[31]. As lower expression of KLHL17 is associated with increased risk, this suggests that higher vimentin and nestin expression are associated with an increased risk of PDAC, likely pro-moting inflammation, ADM and EMT.

Unfortunately, our in vitro model system is limited with regards to understanding the effects of inflammation, ADM, and EMT in the context of KLHL17 expression in the pancreas in vivo. Most pancreas cell lines are derived from tumors or metastatic lesions and have ductal characteristics. Available human normal-derived pancreatic cell lines are duct epithelial cells and would not recapitulate ADM under such stimuli. Primary human pancreas cells, particularly acinar cells, are difficult to maintain in culture as the acinar cell population is quickly lost due to ADM and cell death[60]. Due to these limitations, additional studies are needed to investigate the role of KLHL17 in the context of pancreatic injury, inflammation, and pancreatic tissue.

Taken together, we propose an explanatory model for the chr1p36.33 PDAC GWAS locus wherein lower KLHL17 expression allows for unresolved inflammation and acinar cell injury resulting in increased likelihood of the development of PanINs and progression to PDAC. Under cellular stress, JunB and JunD are induced by extracellular stimuli, such as cytokines[61]. Under such conditions, the preferential binding of JunB/D to the protective allele at rs13303160 may drive KLHL17 expres-sion, resulting in the recognition of vimentin and nestin for ubiquitina-tion and degradation and mitigation of ADM, EMT and associated inflammation. In contrast, when the risk allele is present, JunB/D may not sufficiently induce KLHL17 expression to resolve inflammation (Fig. 6d). Further in vivo studies are needed to fully explore the role of KLHL17 in the pancreas especially in modulating carcinogenesis.

## Methods

### Ethics
All studies obtained consent from participants and Institutional Review Board (IRB) approvals including IRB certifications permitting data sharing in accordance with the NIH Policy for Sharing of Data Obtained in NIH Supported or Conducted Genome-Wide Association Studies (GWAS). Additionally, the PanScan study was approved by the NCI Special Studies Institutional Review Board. All methods and pro-cedures followed the international criteria outline in the Declaration of Helinski.

### UK Biobank PDAC GWAS and meta-analysis
We obtained the cancer registry and hospital inpatient information phenotype data from UKBB on August 6th, 2021 (Approval #29565, Laufey T. Amundadottir). Case criteria for patients diagnosed with pancreatic ductal adenocarcinoma (PDAC): [22006-0.0] = 1 (white British), [22010-0.0] ≠ 1 (recommended genomic analysis exclusions), [22019-0.0] ≠ 1 (sex chromosome aneuploidy exclusion), [31-0.0] (self-reported sex; 0 = female, 1 = male) is consistent with [22001-0.0] (genetic sex), [40012] (Tumor behavior) include only: '3'—malignant,

primary site, [40008] (age at cancer diagnosis), [40006]—Type of Cancer ICD10 codes (C25*—except for C25.4). Control criteria: exclude control if any of the following Data Fields have values > 0, [134]—Number of self-reported cancers, [2453]—Cancer diagnosed by doctor; exclude if any ICD10 codes in the following starting with C*, [40006]—Type of Cancer ICD10 codes, [41202]—Diagnoses—main ICD10 summary information, [41204]—Diagnoses—secondary ICD10 summary information, [41270] Diagnoses—ICD10 from Hospi-tal inpatient information; exclude if any question in the following with a value that is not "NA", [20001]—Cancer code self-reported, [20007] —Interpolated age when cancer first occurred, [40001]—Underlying primary cause of death ICD10; [40008]—Age at cancer diagnosis, [40011]—Histology of cancer tumor, [40012]—Behavior of cancer tumor, [40013]—Type of cancer ICD9 codes, [84]—Cancer year/age first occurred (Medical conditions), [40009]—Reported occurrences of cancer—Cancer Register; [31-0.0] (self-reported sex) is consistent with [22001-0.0] (genetic sex); [22010]—Recommended genomic analysis exclusions—filter samples based on poor heterozygosity/ missingness as per UKB analysis (exclude if "1"); [22018] - Genetic Relatedness exclusions - exclude those with "1" or "2"; [22019]—Sex chromosome aneuploidy—those with 'Yes' were excluded; [22021]— Genetic kinship to other participants—we only used those with "No kinship found". We selected up to 10 controls for each case based on similar age groups (same gender and birthyear ±5 years) and genetic background (using PCAmatchR[62]). The imputed genotype data were downloaded from UKBB (November 2021). We removed variants with MAF < 0.5%, INFO < 0.3, and completion rates > 10%. Genome-wide association tests were performed using the "Frequentist" additive model in SNPTEST(v2.5.4-beta3)[63] with covariates (age, gender, sig-nificant principal components, and array type).

Summary statistics from four previously published GWAS phases (PanScan I-III[5–8] and PanC4[9], and the UK Biobank summary statistics described above (case subjects: $n = 10,106$ and control subjects: $n = 21,895$) were used for the meta-analysis. The meta-analysis was performed using Metal (03/25/2011)[64].

### Fine-mapping of the GWAS signal
The chr1p36.33 GWAS region was fine-mapped using the chi-squared likelihood ratio test of the GWAS P-values for all SNPs. Linkage dis-equilibrium (LD) with the tag SNP rs13303010 was determined for every GWAS variant between chr1:1-2,300,000 (hg19) using LDLink[65] European population. Variants with a LLR < 1:100 and an LD $r^2 > 0.8$ were considered likely functional SNPs for experimental validation. Additionally, the Sum of Single Effects (SuSiE v.012.35 in RStudio 2022.02.3 + 492), a Bayesian approach, was used to identify credible sets of variants likely harboring functional variant(s)[13]. A threshold of 0.9 probability that the credible set of variants contains a causal SNP and $L = 10$ (up to 10 variants in a credible set) was used. Plink v1.07 and PanScan III data (including cases and controls) was used to generate the required LD matrix[7,8].

### Electrophoretic mobility shift assays (EMSA)
SNP-centered 31 bp oligonucleotides (Supplementary Table 2) were labeled with IRDye©700 fluorescent dye on the 5′ end and HPLC purified (Integrated DNA Technologies, Coralville, IA). Competitor oligonucleotides were unlabeled (IDT). Oligonucleotides were annealed at 99 °C and cooled slowly to room temperature. EMSA binding reactions were as follows: 10X binding buffer, 50% glycerol, polyDiDC, nuclear lysate (2.5–5 μg, ActiveMotif, Carlsbad, CA), labeled oligo (5 nM), water. Competition binding reactions included unlabeled oligos at 50X and 100X of the labeled oligo concentration. Reactions were incubated at room temperature in the dark for 20 min. Reactions were then loaded on 4–12% gradient TBE gels (Invitrogen, Waltham, MA) with 0.5X TBE and ran for 100 min at 90 V. Gels were imaged on the BioRad ChemiDoc™ with the IR680 setting. For the TPA-stimulated

EMSAs, 2-h TPA stimulated HeLa nuclear extract (ActiveMotif, Carlsbad, CA) was used.

For supershift assays, the nuclear lysate and antibodies for ELF2, JunB or JunD were incubated for 20 min at room temperature prior to the addition of other binding reaction components. The complete reactions were incubated for an additional 20 min.

For EMSAs with recombinant proteins, 135 ng or 270 ng of recombinant protein was used in place of nuclear lysate. Recombinant ELF1 (TP760629), ELF2 (TP760288), ELF3 (TP300631), ELF4 (TP761826), JUNB (TP303595), JUND (TP316958 4), c-FOS (TP760257), FOS1L (TP302104), FOSB (TP762032), FOS2L (TP760114) proteins were purchased from Origene (Rockville, MD). c-JUN was purchased from Abcam (Waltham, MA) (ab84134). ELF5, which has negligible expression in the pancreas, was not tested.

## Plasmids
Luciferase backbone plasmids used for reporter assays were pGL4.23 and pGL4.14 (Promega, Madison, WI). Four gene blocks (forward orientation reference allele, forward orientation alternate allele, reverse orientation reference allele, reverse orientation alternate allele) of varying sizes (141-201 bps) were synthesized for rs13303160, rs13303010, and rs7524174 (Integrated DNA Technologies, Coralville, IA). Sequences to be assayed (Supplementary Table 3) were determined based on ATAC-seq and ChromHMM annotations previously generated in PDAC cell lines[24] with the goal of being inclusive of regulatory elements as well as sequence complexity. A 165 bp sequence for rs13303327 was cloned from a heterozygous HapMap CEPH subject (NA12716) (Primers in Supplementary Table 3). The gene blocks were ligated into the luciferase plasmid as either enhancers or promoters. *KLHL17* pcDNA3 plasmid was purchased from GenScript (Piscataway, NJ) and subcloned into the pFUGW with a TREG3 promoter for tetracycline-inducible lentiviral expression (Frederick National Cancer Research Laboratories). For CRISPRi experiments, stably expressing dCas9-KRAB-ZIM3 cell lines were generated using pLX303-ZIM3-KRAB-dCas9 (Addgene # 154472, Watertown, MA). Guide RNAs were purchased from Integrated DNA Technologies (Coralville, IA) and cloned into pU6-sgRNA EF1Alpha-puro-T2A-BFP (Addgene #60955, Watertown, MA #60955). Guide RNA sequences (Supplementary Table 4) for CRISPRi were determined using the UCSC Genome Browser CRISPR targets track that uses the CRISPOR program[66]. The negative targeting control (sgNegative) is targeted to an open chromatin region within the same topologically associated domain as *KLHL17*. Sequences can be found in Supplementary Table 4. siRNA pools for *NOC2L* (L-020539-02-0010, Horizon Discovery Dharmacon Lafayette, CO), *KLHL17* (L-031770-00-0020, Horizon Discovery Dharmacon Lafayette, CO) were purchased.

## Cell culture
Cell lines (MIA PaCa-2 (CRM-CRL-1420), PANC-1 (CRL-1469), Hs766T (HTB-134), and HEK293T (CRL-3216)) were purchased from ATCC (Manassas, VA). SW1990 cell line was a gift from Dr. Jaiswal Kshama. PANC-1, Hs766T, and HEK293T cells were maintained in DMEM with 10% FBS. MIA PaCa-2 cells were maintained in DMEM with 2.5% horse serum and 10% FBS. SW1990 cells were maintained in RPMI and 10% FBS. All cells were grown at 37 °C with 5% $CO_2$. For virus production, HEK293T cells were grown in high-glucose DMEM media supplemented with 10% FBS, 1% glutamine, and 1% sodium pyruvate. Cell lines were routinely tested for mycoplasma and were always found to be negative. Cell lines were also tested for authentication with a panel of short tandem repeats (STRs) via the Identifiler kit (Life Technologies, Carlsbad, CA) and compared with ATCC and DSMZ (German Collection of Microorganisms and Cell Cultures–https://www.dsmz.de/) STR profile datasets. All cell lines with profiles in the databases matched and those not with profiles in this database matched earlier passages of these cell lines in use in our laboratory. Commonly misidentified cell lines were not used in this study.

## Lentivirus production
HEK293T cells were plated 24 h prior to transfection. The plasmid of interest and packaging vectors (psPAX2 Addgene #12260, pMD2.G Addgene #12259) were transfected using Lipofectamine 3000 (Thermo Fisher Scientific, Waltham, MA) and media was changed 6−8 h post-transfection. Forty-eight hours later, virus was harvested, filtered with a 0.45-micron syringe filter, and precipitated overnight at 4 °C with 2X PEG precipitation buffer. The precipitated virus was centrifuged at $1000 \times g$ for 30 min, 4 °C, the supernatant was removed and pelleted virus resuspended in PBS.

## Generation of stable cell lines
MIA PaCa-2 and PANC-1 cell lines were used to make stably expressing dCas9-KRAB-Zim3 and doxycycline-inducible KLHL17-FLAG over-expressing lines. For stably expressing dCas9 lines, cells were transduced with the virus for 24 h. After 24 h, selection with 10 μg/mL of blasticidin was initiated. Once selection was complete, cells were diluted in a 96-well plate at a seeding density of 0.5 cells/well to isolate individual cell colonies. These colonies were assessed for dCas9-KRAB-Zim3 protein expression by western blot. Transduction with CRISPRi gRNAs was performed in the same manner using the stably expressing dCas9 cells and selected with 8–10 μg/mL of puromycin. To generate doxycycline-inducible KLHL17-FLAG cell lines, previously generated MIA PaCa-2 and PANC-1 cells stably transduced with the Clontech (Mountain View, CA) TetOn3G transactivator plasmid (pLVX-Tet3G)[67], were transduced with the TREG3p-FUGW-KLHL17 lentiviral expression plasmid (described above) in a 12-well plate. Media was changed 24 h post-transduction and puromycin selection (8-10 μg/mL) initiated at 48 h.

## Luciferase assays
MIA PaCa-2, PANC-1, and HEK293T cells were seeded in 48-well plates 24 h prior to transfection. Cells were co-transfected with 1 μg of the luciferase plasmid pGL4.14 or pGL4.23, (Promega, Madison, WI) and 35 ng of pGL4.74 Renilla vector (Promega, Madison, WI) using Lipofectamine 2000 (Thermo Fisher Scientific, Waltham, MA). Forty-eight hours following transfections, cells were washed twice with PBS. Luciferase assays were performed with the Promega Dual Luciferase Reporter Assay following the manufacturer's instructions. Luciferase activity was normalized to Renilla luciferase activity and reported as the fold-change relative to the empty luciferase vector. A Student's two-tailed t-test was performed to test for statistically significant differences between alleles. For luciferase assays with TPA, cells were simultaneously transfected and stimulated with 200 nM Phorbol 12-myristate 13-acetate (TPA, Millipore Sigma, Burlington, MA) or DMSO 24 h after plating and harvested 48 h after transfection. Luciferase activity was determined as described above. Significance testing was performed on the ratio of A to G relative luciferase activity using an unpaired two-tailed t-test.

## Colocalization analysis
Colocalization analysis was performed on the 2018 GWAS summary statistics[4] and the GTExv7 pancreas eQTL data (downloaded from the GTEx portal in 2019) using coloc (version 4.0)[27]. SNPs within 1 Mb of rs13303010 in the GWAS summary statistics were used. This list of SNPs was used to filter the gene-SNP pairs in the GTEx data and colocalization analysis was performed.

## Cell growth assays
For *KLHL17* overexpression cell proliferation assays, overexpression in stable cell lines was induced with 100 ng/mL of doxycycline 24 h prior

to plating. Cells were then plated in 12-well plates. Twenty-four hours after plating (defined as 0 h), cell images were taken on the Lionheart (Biomek, Brea, CA) plate reader every 48 h for seven days.

For *KLHL17* and *NOC2L* knockdown cell growth assays, cells were plated 24 h prior to transfection in a 6-well plate. Cells were then transfected with siRNA pools against *NOC2L* (L-020539-02-0010, Horizon Discovery Dharmacon Lafayette, CO), *KLHL17* (L-031770-00-0020, Horizon Discovery Dharmacon Lafayette, CO) using RNAiMAX (Thermo Fisher Scientific Waltham, MA non-targeting control (D-001220-01-05, Horizon Discovery Dharmacon Lafayette, CO) using RNAiMAX (Thermo Fisher Scientific Waltham, MA). Cell counts were then taken using the Lionheart (0 h). Counts were taken daily for 7 days. At 72 h, cells were re-transfected with siRNA. For CRISPRi growth assays, cells were plated in a 12-well plate following complete selection with puromycin. The first count was taken 24 h after plating and considered 0 h. Counts were subsequently taken every 24 h for 7 days. For all cell proliferation assays, the cell count for each day was normalized to the 0-h cell count and is represented as a fold change relative to the initial cell count. Significance testing was performed using an unpaired two-tailed t-test on the non-targeting/negative controls and the knockdown at each time point.

### RNA isolation and reverse transcriptase quantitative PCR
RNA was isolated using the QIAGEN RNeasy kit with a DNase digest and the QIAcube (QIAGEN, Germantown, MD). RNA was reverse transcribed to cDNA using SuperScript III Reverse Transcriptase (Invitrogen, Waltham, MA). Gene expression levels were quantified by qRT-PCR using TaqMan (Thermo Fisher Scientific, Waltham, MA) assays: *NOC2L* (Hs00610834_g1), *KLHL17* (Hs00938625_g1), *HPRT* (Hs99999909_m1).

### Chromatin Immunoprecipitation
Chromatin Immunoprecipitation was performed using the ActiveMotif (Carlsbad, CA) High Sensitivity ChIP kit with SW1990 and Hs766T PDAC cell lines. For JunB/D ChIP-qPCR, cell media was replaced approximately 16 h prior with serum free media and cells were stimulated with 200 nM of TPA (Millipore Sigma, Burlington, MA) for 3.5 h prior to cross-linking. Following crosslinking, cells were lysed with a 25-gauge syringe prior to sonication. Samples were sonicated using the Covaris ME220 (Woburn, MA). Shearing efficiency and chromatin concentration was assessed for the input. Immunoprecipitation was set up following the ActiveMotif protocol. Two hundred microliters of sheared chromatin were used for IPs with either ELF2 (4 μg), JunD (4 μg), JunB (10 μL) or IgG (4 μg) antibodies. IPs were incubated at 4 °C overnight, then protein G agarose beads were added for 3 h. Following DNA purification, enrichment of ELF2, JunB, or JunD at the regions of interest was assessed using qPCR (Supplementary Table 5 for primers) using SYBR Green Master Mix. Percent input was calculated following ActiveMotif's protocol using a standard curve from input DNA for each primer set. Allelic enrichment was determined using TaqMan genotyping assays (C_57466801_10 and custom design). The ratio of A to G alleles was calculated and compared to the input DNA allelic ratio. Control regions were determined from ELF2 and JunB ChIP-seq data (GSE177468, GSE119930, respectively). Significance testing was performed between the percent input for IgG and the antibody of interest using an unpaired two-tailed t-test. For the allele-specific enrichment, unpaired two-tailed t-tests were performed on the ratio of A to G in the IP compared to the input.

### Antibodies
Antibodies for EMSA: ELF2 (ab28726, Abcam; 12499-1-AP, Proteintech Rosemont, IL), JunB (C37F9, Cell Signaling Technologies, Danvers, MA), JunD (D17G2, Cell Signaling Technologies, Danver, MA). For Western Blot and Immunoprecipitation: JunB (1:1000, C37F9, Cell Signaling Technologies, Danver, MA), JunD (1:1000, D17G2, Cell Signaling Technologies, Danver,MA), FLAG (1:1000, F1804, Millipore Sigma,

Burlington, MA), KLHL17 (1:500, PA5-56689 Thermo Fisher Scientific, Waltham, MA); 1:500, HPA031251, Millipore Sigma, Waltham, MA), GAPDH (1:1000, ab125247, Abcam, Waltham, MA), SP1 (1:000 ab13370, Abcam); mouse anti-rabbit light chain specific antibody HRP (1:5000, C840Z39 Jackson ImmunoResearch, West Grove, PA); donkey anti-mouse secondary HRP (1:5000, ab7061, Abcam, Waltham, MA), donkey anti-rabbit secondary HRP (1:5000, ab205722, Abcam, Waltham, MA); For ChIP: ELF2 (4 μg,12499-1-AP, Proteintech, Rosemont, IL), JunB (10 μL, C37F9,Cell Signaling Technologies, Danver, MA), JunD (4 μg, 720035, Invitrogen, Waltham, MA), Rabbit IgG (4 μg, 2729S, Cell Signaling Technologies, Danver, MA). Immunofluorescence: KLHL17 (1 μg/mL, HPA031251, Millipore Sigma, Burlington, MA), AlexaFluor647 (1:1000, A-31573, Thermo Fisher Scientific, Waltham, MA)

### Western blot analysis
Cells were lysed with either NP-40-DOC-SDS lysis buffer (150 mM NaCl, 50 mM Tris, 1% NP-40, 1% Sodium Deoxycholate, 1% Sodium dodecyl sulfate) or NE-PER™ Nuclear and Cytoplasmic extraction kit (Thermo Fisher Scientific, Waltham, MA). Lysates were run on Criterion™ XT precast 3-8% Tris-Acetate gels (Bio-Rad, Hercules, CA) using XT running buffer and transferred to a PDVF membrane using a standard wet transfer or Bolt™ 4-12% Bis-Tris Plus gels (Invitrogen, Waltham, MA) using MOPS running buffer and transferred to PDVF membranes using the iBlot™ Transfer Stacks (Invitrogen, Waltham, MA). Membranes were blocked in 5% Bovine Albumin Serum and incubated with primary antibody overnight at 4 °C. The appropriate HRP secondary antibody was added for a 1 h incubation at room temperature. Following washes with TBS-t, chemiluminescence was detected with SuperSignal™ West Femto Maximum Sensitivity Substrate (Thermo Fisher Scientific, Waltham, MA) and imaged on the Bio-Rad ChemiDoc™.

### Immunoprecipitation
Cells were treated with 1 μM of MG-132 proteosome inhibitor (Millipore Sigma, Burlington, MA) 16 h prior to harvest. Cells were harvested and lysed using 50 mM HEPES (pH 7.4), 150 mM NaCl, 0.5 mM EDTA, 0.1% NP-40, protease inhibitor, and 10 μg MG-132 on ice for 5 min then freeze-thawed once. Samples were centrifuged at 14,000 × g for 5 min to pellet cell debris. One milligram of protein extract was incubated with 2 μg of FLAG antibody at 4 °C for 2 h. Protein G beads (Invitrogen, Waltham, MA) were added and incubated for an additional 30 min. Five IPs (for a total of 5 mg of protein used) were combined and washed three times with 50 mM HEPES. Ten percent of the IP was used for western blot analysis and the remaining 90% was subjected to mass spectrometry.

### Protein digestion and TMT labeling
Samples for mass-spectrometry included 3 independent FLAG-KLHL17 IPs, 4 independent Empty Vector FLAG IPs, as a background control, 3 biological replicates of MIA PaCa-2-KLHL17 and PANC-1 KLHL17 cells treated with 0.1, 1, 10, and 100 ng/mL of doxycycline, and 3 biological replicates of CRISPRi sgNeg (control) and sgKLHL17-1 in PANC-1 and MIA PaCa-2 cells at both Day 8 and Day 36. The cell pellets were lysed in EasyPrep Lysis buffer (Thermo Fisher, CA) according to the manufacturer's protocol. Lysates were clarified by centrifugation and protein concentration was quantified using the BCA protein estimation kit (Thermo Fisher, CA). Fifteen micrograms of lysate were reduced, alkylated and digested by the addition of trypsin at a ratio of 1:50 (Promega) and incubating overnight at 37 °C.

For TMT labeling 100 μg of TMTpro label (Thermo Fisher, CA) in 100% ACN was added to each sample. After incubating the mixture for 1 h at room temperature with occasional mixing, the reaction was terminated by adding 50 μL of 5% hydroxylamine, 20% formic acid. The peptide samples for each condition were pooled and peptide clean-up was performed using the proprietary peptide clean up columns from the EasyPEP Mini MS Sample Prep kit (Thermo Fisher, CA).

## High pH reverse phase fractionation

The first dimensional separation of the peptides was performed using a Waters Acquity UPLC system coupled with a fluorescence detector (Waters, Milford, MA) using a 150 mm × 3.0 mm Xbridge Peptide BEM™ 2. 5 μm C18 column (Waters, MA) operating at 0.35 mL/min. The dried peptides were reconstituted in 100 μL of mobile phase A solvent (10 mM Ammonium Formate, pH 9.4). Mobile phase B was 10 mM Ammonium Formate/90% acetonitrile, pH 9.4. The column was washed with mobile phase A for 5 min followed by gradient elution 10–50% B (5–60 min) and 50–75% B (60–70 min). The fractions were collected every minute. These 60 fractions were pooled into 24 fractions. The fractions were vacuum centrifuged to dryness and stored at −80 °C until analysis by mass spectrometry.

## Mass spectrometry acquisition and data analysis

The dried peptide fractions were reconstituted in 0.1% TFA and subjected to nanoflow liquid chromatography (Thermo Easy nLC 1200, Thermo Scientific, Thermo Scientific) coupled to an Orbitrap LUMOS mass spectrometer (Thermo Scientific, CA). Peptides were separated using a low pH gradient using a 5-50% ACN over 120 min in mobile phase containing 0.1% formic acid at 300 nL/min flow rate. MS scans were performed in the Orbitrap analyzer at a resolution of 120,000 with an ion accumulation target set at $4e^5$ and max IT set at 50 ms over a mass range of 400–1600 $m/z$. Ions with determined charge states between 2 and 5 were selected for MS2 scans. A cycle time of 3 s was used and a quadrupole isolation window of 0.7 m/z was used for MS/MS analysis. An Orbitrap at 50,000 resolutions with a normalized AGC set at 250 followed by maximum injection time set as "Auto" with a normalized collision energy setting of 38 was used for MS/MS analysis.

Acquired MS/MS spectra were searched against a human Uniprot protein database using a SEQUEST HT and percolator validator algorithms in the Proteome Discoverer 2.4 software (Thermo Scientific, CA). The precursor ion tolerance was set at 10 ppm and the fragment ions tolerance was set at 0.02 Da along with methionine oxidation included as dynamic modification. Carbamidomethylation of cysteine residues and TMT16 plex (304.2071 Da) was set as a static modification of lysine and the N-termini of the peptide. Trypsin was specified as the proteolytic enzyme, with up to 2 missed cleavage sites allowed. Searches used a reverse sequence decoy strategy to control for the false peptide discovery and identifications were validated using the percolator software. Only peptides with less than 50% co-isolation interference were used for quantitative analysis.

Reporter ion intensities were adjusted to correct for the impurities according to the manufacturer's specification and the abundances of the proteins were quantified using the summation of the reporter ions for all identified peptides. The reporter abundances were normalized across all the channels to account for equal peptide loading. Data analysis and visualization were performed in Microsoft Excel and R.

## On-bead trypsin digestion and LC-MS/MS analysis

Beads were resuspended in 30 μL of 50 mM HEPES (pH 8.0) and heated at 95 °C for 10 min. Samples were treated with 2 μg of trypsin and incubated at 37 °C overnight with constant shaking. The supernatant containing the tryptic digests was collected after centrifugation. The residual beads were washed twice with 50 mM HEPES (pH 8.0), and the supernatant and washes combined for maximum recovery. Peptides were desalted using EasyPep MS sample prep kit (Thermo Scientific, CA) and lyophilized. The dried peptides were suspended in 15 μL of 0.1% TFA and analyzed using an EASY-nLC 1200 (ThermoFisher Scientific, Waltham, MA) in front of an Q Exactive HF (ThermoFisher Scientific, Waltham, MA) equipped with an EasySpray ion source. The desalted tryptic peptide was loaded onto an Acclaim PepMap 100 (75 μM × 2 cm) C18 trap column (ThermoFisher Scientific, Waltham, MA) followed by a separation on PepMap RSLC C18 (75 μM × 25 cm) analytical column. The peptides were eluted with a 5% to 27% gradient

of Acetonitrile with 0.1% Formic acid over 60 min and 27% to 40% gradient of Acetonitrile with 0.1% Formic acid over 45 min with a flow rate of 300 nL/min. The MS1 was performed at 60,000 resolutions over mass range of 380 to 1580 $m/z$, with a maximum injection time of 120 ms and an AGC target of 3e6. The MS2 scans were performed at resolution of 15,000, normalized collision energy set at 27, maximum injection time of 50 ms and an AGC target of 2e5.

MS files were searched with Proteome Discoverer 2.4 using the Sequest node. Data were searched against the Uniprot human database using a full tryptic digest, 2 max missed cleavages, minimum peptide length of 6 amino acids and maximum peptide length of 40 amino acids, an MS1 mass tolerance of 10 ppm, MS2 mass tolerance of 0.02 Da

## Substrate identification

For analysis to identify enriched proteins in the IP mass spectrometry data we used the peptide matching spectral (PSM) counts. We first added a pseudo count of 1 was added to all peptide matching spectral (PSM) counts to adjust for proteins that had 0 PSM counts in order to calculate a fold-enrichment relative to the empty vector IP. The median of the pseudo count transformed PSM counts was calculated ($n = 2$ for KLHL17-FLAG; $n = 3$ for empty vector IP), and the fold change between the KLHL17-FLAG IP/Empty vector IP was calculated. $P$-values were calculated using the PSM + 1 counts. Proteins that were significantly ($P < 0.05$) enriched in the KLHL17 IP over the empty vector IP and had an enrichment fold change > 1.5 were considered candidate substrates. We then compared this list to the enriched proteins identified in the pilot IP-MS experiment. The overlapping proteins ($n = 62$) were moved forward and cross-referenced with the global proteomics experiment examining protein expression with different levels of KLHL17 expression. Proteins that were enriched and displayed a doxycycline dose-dependent decrease in protein expression were determined to be likely KLHL17 substrates.

## Immunofluorescence analysis

MIA PaCa-2 and PANC-1 cells overexpressing *KLHL17* were plated on coverslips and induced with 100 ng/mL of doxycycline for 72 h. Cells were then washed with PBS and fixed with 4% paraformaldehyde (Thermo Fisher Scientific, Waltham, MA) for 15 min. Following fixation, cells were permeabilized with 0.25% Triton-X for 15 min. Samples were then blocked in 2% BSA (Millipore Sigma, Burlington, MA) for 1 h at room temperature. Primary KLHL17 antibody was added to the cells at 1 μg/mL and incubated at 4 °C overnight. Cells were washed with PBS and incubated with an AlexaFluor 647 secondary antibody for 1 h at room temperature in the dark. Coverslips were then mounted on slides using ProLong™ Diamond Antifade Mountant with DAPI (Thermo Fisher Scientific, Waltham, MA) and allowed to cure for at least 24 h at 4 °C in the dark. Slides were imaged with a Zeiss Microscope.

## In silico knockdown and pathway analysis

An in silico knockdown analysis for *KLHL17* was performed using GTExv8 normal pancreas tissue sample-derived RNA-seq data (downloaded from the GTEx portal in 2020) as previously described[28]. Briefly, we scaled GTExv8 pancreas gene expression counts to account for sequencing depth and RNA composition across all samples ($n = 328$) to give normalized counts of the trimmed mean of M-values (TMM) using EdgeR (3.38.1 in RStudio 2022.02.3 + 492)[37]. Genes with no reads for > 20% of the samples were excluded. Normalized reads were used to segregate samples into quartiles based on *KLHL17* expression. Only the samples in the top and bottom quartiles (n = 82 each) were used for downstream analysis. The raw counts for these selected samples were re-normalized for sequencing depth to obtain pseudo-counts which were analyzed using the quantile-adjusted conditional maximum likelihood (qCML) method in EdgeR. Differential expression (log2 [bottom/top quartile] and $P$ values) was then assessed using an exact

test. The statistically significant (FDR < 0.05) differentially expressed genes were subjected to Gene Set Enrichment Analysis (GSEA) using webgestalt.org[39]. The ranked list for the GSEA was based on log2 fold change. Without an FDR filter, there was no significant enrichment of gene sets. For Ingenuity Pathway Analysis, a FDR < 0.05 and a log2 fold change > |0.5| was used to filter genes for input (IPA, QIAGEN, Germantown, MD). For IPA, both the log2 fold change and FDR were considered in the analysis of enriched pathways.

### Transcription factor binding prediction
In silico Transcription Factor binding prediction was performed using PrEdict Regulatory Functional Effect of SNPs by Approximate *P* value Estimation (PERFECTOS-APE; https://opera.autosome.org/perfectosape/). Briefly, the SNPs with allele-preferential activity were submitted for analysis to determine the probability of a TF binding site from the position matrices in the HOCOMOCO11 transcription factor database. Once a *P*-value of predicted TF binding sequence was determined for each allele, a fold change was calculated[25].

### Reporting summary
Further information on research design is available in the Nature Portfolio Reporting Summary linked to this article.

## Data availability
The proteomics data generated in this study have been deposited in the MassIVE database under accession code #MSV000096025 [https://massive.ucsd.edu/ProteoSAFe/private-dataset.jsp?task=e357b08275e34f89b280e716ccd20d8f]. The PDAC GWAS data used in this study are available under controlled access because of data use limitations and can be requested through dbGaP: phs000206.v6.p3 [https://www.ncbi.nlm.nih.gov/projects/gap/cgi-bin/study.cgi?study_id=phs000206.v6.p3] and phs000648.v1.p1 [https://www.ncbi.nlm.nih.gov/projects/gap/cgi-bin/study.cgi?study_id=phs000648.v1.p1]. UK Biobank data is available through the UK Biobank (https://www.ukbiobank.ac.uk/). Source data are provided with this paper.

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

## Acknowledgements

This study utilized the high-performance computational capabilities of the Biowulf Linux cluster at the NIH, Bethesda, MD, USA (http://biowulf.nih.gov). The authors would like to thank the Frederick National Cancer Research Laboratories for the generation of inducible plasmids. We also thank participants and clinical coordinators participating in the Pancreatic Cancer Cohort Consortium, Pancreatic Cancer Case-Control Consortium and the UK Biobank for providing samples for the GWAS studies. The authors acknowledge the research contributions of the Cancer Genomics Research Laboratory for their expertise, execution, and support of this research in the areas of project planning, wet laboratory processing of specimens, and bioinformatics analysis of generated data. The data used for the analyses described in this manuscript were obtained from the GTEx Portal versions 7 and 8 pancreas data in 2019 and 2020. This research has been conducted using data from UK Biobank (Approval #29565, Laufey T. Amundadottir), a major biomedical database (www.ukbiobank.ac.uk)[20]. This work was

supported by the Intramural Research Program (IRP) of the Division of Cancer Epidemiology and Genetics (DCEG), National Cancer Institute (NCI), US National Institutes of Health (NIH). This project has been funded in whole or in part with Federal funds from the National Cancer Institute, National Institutes of Health, under NCI Contract No. 75N910D00024 (L.T.A.). The content of this publication does not necessarily reflect the views or policies of the Department of Health and Human Services, nor does mention of trade names, commercial products, or organizations imply endorsement by the U.S. Government. The American Cancer Society (ACS) funds the creation, maintenance, and updating of the Cancer Prevention Study II cohort. The authors express sincere appreciation to all Cancer Prevention Study-II participants, and to each member of the study and biospecimen management group. The authors would like to acknowledge the contribution to this study from central cancer registries supported through the Centers for Disease Control and Prevention's National Program of Cancer Registries and cancer registries supported by the National Cancer Institute's Surveillance Epidemiology and End Results Program. The American Cancer Society funds the creation, maintenance, and updating of the Cancer Prevention Study-II cohort (and/or Cancer Prevention Study-3). The authors express sincere appreciation to all Cancer Prevention Study-II participants, and to each member of the study and biospecimen management group. The authors would like to acknowledge the contribution to this study from central cancer registries supported through the Centers for Disease Control and Prevention's National Program of Cancer Registries and cancer registries supported by the National Cancer Institute's Surveillance Epidemiology and End Results Program. Where authors are identified as personnel of the International Agency for Research on Cancer/World Health Organization, the authors alone are responsible for the views expressed in this article and they do not necessarily represent the decisions, policy or views of the International Agency for Research on Cancer / World Health Organization. We acknowledge funding for the Women's Health Study (WHS) source of data: CA047988, CA182913, HL043851, HL080467, and HL099355. We acknowledge WHI investigators listed here: https://www-whi-org.s3.us-west-2.amazonaws.com/wp-content/uploads/WHI-Investigator-Short-List.pdf. The WHI program is funded by the National Heart, Lung, and Blood Institute, National Institutes of Health, U.S. Department of Health and Human Services through 75N92021D00001, 75N92021D00002, 75N92021D00003, 75N92021D00004, 75N92021D00005. The EPIC-Norfolk study (https://doi.org/10.22025/2019.10.105.00004) has received funding from the Medical Research Council (MR/N003284/1, MC-UU_12015/1 and MC_UU_00006/1) and Cancer Research UK (C864/A14136). We are grateful to all the participants who have been part of the project and to the many members of the study teams at the University of Cambridge who have enabled this research. Support for title page creation and format was provided by AuthorArranger, a tool developed at the National Cancer Institute.

## Author contributions

K.E.C. and L.T.A. conceived, designed, and oversaw the study. K.E.C., K.H., E.A., I.C. performed the experiments. K.E.C., K.H., and E.A. performed data analysis. S.D., G.D., and T.A. performed the proteomics experiments and provided expertise for experiment design and data analysis. J.Z., D.R.E., A.O.B., and J.W.H. assisted with bioinformatics and statistical analysis of the GWAS, ChromHMM, and fine-mapping. L.T.A., S.J.C., R.Z.S., B.M.W., A.P.K., J.P.S., and authors from the Pancreatic Cancer Cohort Consortium and Pancreatic Cancer Case-Control Consortium provided the samples, analysis, and oversaw the GWAS. K.E.C. and L.T.A. wrote the manuscript. All authors reviewed the manuscript.

## Funding

## Competing interests

The authors declare no competing interests.

## Additional information

## Pancreatic Cancer Cohort Consortium

Jun Zhong[1], Demetrius Albanes[4], Gabriella Andreotti[9], Alan A. Arslan[10], Laura Beane-Freeman[9], Sonja I. Berndt[9], Julie E. Buring[11,12], Daniele Campa[13], Federico Canzian[14], Stephen J. Chanock [3], Yu Chen[15], Charles C. Chung[16], A. Heather Eliassen[12,17,18], J. Michael Gaziano[11,19,20], Edward L. Giovannucci[12,18], Phyllis J. Goodman[21], Christopher A. Haiman[22], Belynda Hicks[16], Amy Hutchinson[16], Miranda R. Jones[23], Verena Katzke[24], Charles Kooperberg[25], Peter Kraft[26], I-Min Lee[11,12], Loic LeMarchand[27], Núria Malats[28,29], Michelle R. Manning[16], Satu Männistö[30], Roger Milne[31,32,33], Steven C. Moore[4], Lorelei Mucci[12], Alpa V. Patel[34], Ulrike Peters[25], Francisco X. Real[29,35,36],

Nathaniel Rothman[9], Howard D. Sesso[12,19], Veronica W. Setiawan[37], Xiao-Ou Shu[38], Debra Silverman[9], Meir J. Stampfer[12,17,18], Melissa C. Southey[31,32,39], Geoffrey S. Tobias[40], Therese Truong[41], Caroline Um[34], Kala Visvanathan[5,23], Nicolas Wentzensen[42], Emily White[25,43], Chen Yuan[7], Wei Zheng[38], Jean Wactawski-Wende[44], Walter C. Willett[12,18], Brian M. Wolpin [7], Rachael Z. Stoltzenberg-Solomon[4] & Laufey T. Amundadottir [1]✉

[9]Occupational and Environmental Epidemiology Branch, Division of Cancer Epidemiology, National Cancer Institute and Genetics, Rockville, MD, USA. [10]Departments of Obstetrics and Gynecology and Population Health, NYU Grossman School of Medicine, NYU Perlmutter Comprehensive Cancer Center, New York, NY, USA. [11]Division of Preventive Medicine, Department of Medicine, Brigham and Women's Hospital, Boston, MA, USA. [12]Department of Epidemiology, Harvard T.H. Chan School of Public Health, Boston, MA, USA. [13]Unit of Genetics, Department of Biology, University of Pisa, Pisa, Italy. [14]Genomic Epidemiology Group, German Cancer Research Center (DKFZ), Heidelberg, Germany. [15]Department of Population Health, NYU Grossman School of Medicine, NYU Perlmutter Comprehensive Cancer Center, New York, NY, USA. [16]Cancer Genomics Research Laboratory, Frederick National Lab for Cancer Research, Frederick, MD, USA. [17]Channing Division of Network Medicine, Department of Medicine, Brigham and Women's Hospital and Harvard Medical School, Boston, MA, USA. [18]Department of Nutrition, Harvard T. H. Chan School of Public Health, Boston, MA, USA. [19]Division of Aging, Brigham and Women's Hospital, Boston, MA, USA. [20]Boston VA Healthcare System, Boston, MA, USA. [21]SWOG Statistical Center, Fred Hutchinson Cancer Center, Seattle, WA, USA. [22]Department of Preventive Medicine, Keck School of Medicine, University of Southern California, Los Angeles, CA, USA. [23]Department of Epidemiology, Johns Hopkins School of Public Health, Baltimore, MD, USA. [24]Division of Cancer Epidemiology, German Cancer Research Center (DKFZ), Heidelberg, Germany. [25]Division of Public Health Sciences, Fred Hutchinson Cancer Center, Seattle, WA, USA. [26]Trans-Divisional Research Program, Division of Cancer Epidemiology, National Cancer Institute and Genetics, Rockville, MD, USA. [27]Cancer Epidemiology Program, University of Hawaii Cancer Center, Honolulu, HI, USA. [28]Genetic and Molecular Epidemiology Group, Spanish National Cancer Research Center (CNIO), Madrid, Spain. [29]CIBERONC, Madrid, Spain. [30]Department of Public Health, Finnish Institute for Health and Welfare (THL), Helsinki, Finland. [31]Cancer Epidemiology Division, Cancer Council Victoria, East Melbourne, VIC, Australia. [32]Precision Medicine, School of Clinical Sciences at Monash Health, Monash University, Clayton, VIC, Australia. [33]Centre for Epidemiology and Biostatistics, Melbourne School of Population and Global Health, The University of Melbourne, Parkville, VIC, Australia. [34]Department of Population Science, American Cancer Society, Atlanta, GA, USA. [35]Epithelial Carcinogenesis Group, Molecular Oncology Programme, Spanish National Cancer Research Center (CNIO), Madrid, Spain. [36]Universitat Pompeu Fabra, Barcelona, Spain. [37]Department of Population and Public Health Sciences, Keck School of Medicine, University of Southern California, Los Angeles, CA, USA. [38]Division of Epidemiology, Department of Medicine, Vanderbilt Epidemiology Center, Vanderbilt-Ingram Cancer Center, Vanderbilt University School of Medicine, Nashville, TN, USA. [39]Department of Clinical Pathology, The University of Melbourne, Melbourne, VIC, Australia. [40]Division of Cancer Epidemiology National Cancer Institute and Genetics, Rockville, MD, USA. [41]Paris-Saclay University, UVSQ, Inserm, Gustave Roussy, CESP, Villejuif, France. [42]Clinical Genetics Branch, Division of Cancer Epidemiology, National Cancer Institute and Genetics, Rockville, MD, USA. [43]Department of Epidemiology, Fred Hutchinson Cancer Center, Seattle, WA, USA. [44]Department of Epidemiology and Environmental Health, University of Buffalo, Buffalo, NY, USA.

## Pancreatic Cancer Case-Control Consortium

Samuel O. Antwi[45], Paige M. Bracci[46], Steven Gallinger[47], Michael Goggins[6], Manal Hassan[48], Elizabeth A. Holly[46], Rayjean J. Hung[47], Donghui Li[48], Núria Malats[28,29], Rachel E. Neale[49], Kari G. Rabe[50], Harvey A. Risch[51], Herbert Yu[52] & Alison P. Klein [5,6]

[45]Department of Quantitative Health Sciences, Mayo Clinic College of Medicine, Jacksonville, FL, USA. [46]Department of Epidemiology and Biostatistics, University of California San Francisco, San Francisco, CA, USA. [47]Lunenfeld-Tanenbaum Research Institute, Sinai Health System and University of Toronto, Toronto, Canada. [48]Department of Gastrointestinal Medical Oncology, University of Texas MD Anderson Cancer Center, Houston, TX, USA. [49]Population Health Program, QIMR Berghofer Medical Research Institute, Brisbane, Australia. [50]Department of Quantitative Health Sciences, Mayo Clinic College of Medicine, Rochester, MN, USA. [51]Department of Chronic Disease Epidemiology, Yale School of Public Health, New Haven, CT, USA. [52]Epidemiology Program, University of Hawaii Cancer Center, Honolulu, HI, USA.

