## [Transparent Peer Review file · Nature Communications]

Allelic effects on *KLHL17* expression underlie a pancreatic cancer genome-wide association signal at chr1p36.33

Corresponding Author: Dr Laufey Amundadottir

Version 0:

Reviewer comments:

Reviewer #1

(Remarks to the Author)

Connelly et al have clearly expressed the importance and the primary objective of their work. Characterization of the molecular/cellular impact of GWAS signals in the context of disease/phenotype is a major goal of the genomics and biomedical research community.

In this study, Connelly et al, prioritize and functionally characterized fine-mapped SNPs associated with pancreatic ductal adenocarcinoma. The authors convincingly show allele-specific differential TF binding and gene regulatory activity. However, the experiments on *KLHL7* expression levels and cancer-related phenotypes are less conclusive and convincing.

The background section is appropriate and up to date. If possible, I recommend adding information about efforts/methods/findings addressing the role of non-coding variants in disease/cancer.

Materials and methods are reported in sufficient detail and clarity to replicate and reproduce the study.

The purpose and approach of each experiment are clearly described.

Appropriate controls were included in the experiments.

Figures and tables were clearly presented and described throughout the text.

Statistical tests and reporting for fine-mapping and gene-reporter assays are described and appropriate. I could not find description of statistical analysis for the ChIP-qPCR and the qPCR on the CRISPRi cell lines. Please add or make it more explicit.

Are the interpretation of results and study conclusions supported by the data and the study design?

Interpretation of TF allele-specific binding and allele-specific gene regulation activity are supported by the the results.

Can the authors comment on the *KLHL17* cellular localization experiment which is in disagreement with a previous report? Was it the same cell line/type?

Pg5Ln218. I suggest modifying the header '*KLHL17* over-expression and knockdown does not influence cell growth' since the *KLHL7* knockdown attempts were unsuccessful, as stated by the authors.

The authors hypothesize that in the PDAC model *KLHL7* recruits pro-inflammatory proteins for degradation. In addition to the members of the Cullin3-E3 complex (fig 5c), were any peptides from known 'pro-inflammatory' genes detected in the mass spectrometry experiments?

The authors addressed the limitations of the pancreatic cells lines. The authors should also highlight some of the limitations of the in silico *KLHL17* knockdown results and their interpretation.

Reviewer #2

(Remarks to the Author)

In this manuscript, Connelly and colleagues conduct functional dissection of one GWAS risk locus for Pancreatic Cancer. From the tag SNP they first identify additional credible causal SNPs through fine mapping and a Bayesian approach (SuSiE). This is followed by analysis of allele-specific activity using EMSAs, luciferase reporter assays and eQTL analysis. This analysis narrowed it down to three variants and additional analysis of transcription factor binding indicated that rs13303160 represents a functional variant at the locus. They further identify KLHL17 as the likely target gene. This work is timely and is likely to be of interest to a wide community of investigators in functional genomics in cancer.

The introduction is very brief and somewhat disappointing as it fails to put the work in proper context. For example, the reader is not informed whether this is the first GWAS risk locus of pancreatic cancer that has been functionally dissected or there were many other analyzed before. Are there common SNPs between GWAS of PDAC risk and GWAS of traits associated with PDAC risk (obesity, pancreatitis, etc)? In loci identified in previous meta-analyses, was there anything particularly interesting found? Enrichment of pathways?

The work is of high quality and the authors are transparent about the technical difficulties to undertake this type of analysis, especially for such a gene-dense and complex locus. The experiments are conducted with rigor and the interpretation is balanced. The potential chain of causality is well laid out by the authors using assays that are compelling. However, although they identify the likely target gene and propose a plausible scenario for the role of KLHL17 in pancreatic cancer, it is largely speculative. In the absence of additional data tying the role of KLHL17 in inflammation, this excellent paper falls short of being a truly significant advance.

Version 1:

Reviewer comments:

Reviewer #1

(Remarks to the Author)

The authors have satisfactorily addressed my concerns in the revised manuscript. The authors conclusions are supported by the data provided.

Minor comment: pg3ln32: should read "... demonstrating an allele-preferential ...".

Reviewer #2

(Remarks to the Author)

Connelly et al have addressed all the points and concerns raised. I am satisfied with the revised version presented.

Responses to Reviewers' Questions

We thank all reviewers and the editor for their review and constructive comments. We have listed all responses in blue text. Underlined text indicates additions to the resubmitted manuscript (this is also shown in track changes in the manuscript).

Reviewer #1 (Remarks to the Author):

Connelly et al have clearly expressed the importance and the primary objective of their work. Characterization of the molecular/cellular impact of GWAS signals in the context of disease/phenotype is a major goal of the genomics and biomedical research community.

In this study, Connelly et al, prioritize and functionally characterized fine-mapped SNPs associated with pancreatic ductal adenocarcinoma. The authors convincingly show allele-specific differential TF binding and gene regulatory activity. However, the experiments on KLHL7 expression levels and cancer-related phenotypes are less conclusive and convincing.

The background section is appropriate and up to date. If possible, I recommend adding information about efforts/methods/findings addressing the role of non-coding variants in disease/cancer.

Thank you for this recommendation. We have added two additional paragraphs to the introduction with more information highlighting the importance of functional characterization of non-coding germline variants in disease, the genomic and statistical methods used to aid in uncovering the role of non-coding variants in disease, and the PDAC risk loci that have been functionally characterized to date. The new text (page 2) is as follows:

"GWAS have been instrumental in estimating disease risk, identifying candidate genes, and uncovering novel pathways underlying disease development. However, functional characterization of GWAS loci is critical to pinpoint the biological mechanism underlying risk. Most GWAS signals map to non-coding, regulatory regions of the genome and are hypothesized to influence disease risk through allele specific changes in gene expression.¹⁰ Further, each locus is decorated with tens to hundreds of highly correlated variants complicating the identification of functional variant(s) and gene(s) underlying the risk signal. Statistical fine mapping and genomic assays (e.g., ATAC-seq and massively parallel reporter assays) have been beneficial for reducing the number of candidate functional variants to move forward for testing. Chromatin capture and expression quantitative trait locus (eQTL) analysis are valuable for identifying putative functional genes. While these methods greatly assist in the process of functionally characterizing GWAS risk loci, it is still a time-consuming process. In fact, the biological mechanisms underlying only a handful of the 22 published PDAC risk signals have been functionally characterized to date: 5p15.33/*TERT*,¹¹ 16q23.1/*CTRB2*,¹² and 13q22.1/*DIS3*.¹³

Interestingly, some overlap has been observed between GWAS loci for PDAC and associated epidemiological risk factors. A number of PDAC risk loci have common SNPs or colocalize with GWAS for traits that influence PDAC risk: 1q32.1/*NR5A2/T2D*, 2p13.3/*ETAA1/T2D*/waist-hip-ratio (obesity measure), 8q24.21/*MIR1208/T2D*, 9q34/*ABO/T2D*/body fat percentage, 12q24.31/*HNF1A*/Maturity-onset of Diabetes in the Young/*T2D*, 16q23.1/*BCAR1/T2D*, 18q21.32/*GRP/T2D*/BMI/obesity (<https://mvp-ukbb.finngen.fi/> and NHGRI GWAS Catalog¹⁴). Further pathway enrichment analysis of genes +/- 100kb of PDAC GWAS risk loci indicate an enrichment of genes associated with Maturity-onset diabetes of the young (KEGG), Sequence-specific DNA-binding transcription factor activity (GO Molecular Function), Cellular response to UV (GO Biological Process).⁴ Further, DEPICT enrichment analysis indicated that genes associated with GWAS risk loci are highly expressed in numerous gastrointestinal tissues.^{4"}

Materials and methods are reported in sufficient detail and clarity to replicate and reproduce the study.

The purpose and approach of each experiment are clearly described.

Appropriate controls were included in the experiments.

Figures and tables were clearly presented and described throughout the text.

Statistical tests and reporting for fine-mapping and gene-reporter assays are described and appropriate. I could not find description of statistical analysis for the ChIP-qPCR and the qPCR on the CRISPRi cell lines. Please add or make it more explicit.

Thank you for your careful examination of our methods and analyses. All the statistical tests for these experiments were unpaired, two-tailed t-tests. We have updated the text to include the statistical analyses performed for the ChIP-qPCR and CRISPRi qPCR data (please see these updates in the methods section under each respective method).

Are the interpretation of results and study conclusions supported by the data and the study design?

Interpretation of TF allele-specific binding and allele-specific gene regulation activity are supported by the the results.

Can the authors comment on the KLHL17 cellular localization experiment which is in disagreement with a previous report? Was it the same cell line/type?

We thank you for bringing this up as we did not elaborate in the initial submission. The previous report mentioned in the text refers to what is reported on the Human Protein Atlas website (HPA). HPA utilized three cell lines, U251-MG (glioblastoma), A-439 (epidermoid carcinoma), and U2OS (sarcoma), none of which are pancreas cells. All the cell lines in HPA indicated nuclear/nucleoplasm localization. Using the same antibody as HPA, we observe both nuclear and cytoplasmic localization in our MIA PaCa-2 and PANC-1 PDAC overexpression cell lines. It is possible that KLHL17 localization could be cell-type and/or context specific. We have updated the results section of the manuscript (page 5) with the following to indicate these details and differences:

“In the MIA PaCa-2 and PANC-1 overexpressing cell lines, KLHL17 localized throughout the cells (Fig. 5b). This is in contrast to findings from the Human Protein Atlas in A-431 (epidermoid carcinoma), U-251MG (glioblastoma), and U2OS (osteosarcoma) cells, which indicate localization in the nucleoplasm and nuclear bodies.²⁶ suggesting possible cell-type and/or context-specific functions for KLHL17.”

Pg5Ln218. I suggest modifying the header 'KLHL17 over-expression and knockdown does not influence cell growth' since the KLHL7 knockdown attempts were unsuccessful, as stated by the authors.

You raise an important point. We have modified the header to: “Assessment of cell growth after KLHL17 over-expression and knockdown”. If the reviewer and editor prefer a different header, we are open to a change.

The authors hypothesize that in the PDAC model KLHL7 recruits pro-inflammatory proteins for degradation. In addition to the members of the Cullin3-E3 complex (fig 5c), were any peptides from known 'pro-inflammatory' genes detected in the mass spectrometry experiments?

This is a good point. To address this question, we used our immunoprecipitation and global mass spectrometry data together to point us to candidate substrates for KLHL17. We started by analyzing our immunoprecipitation mass-spectrometry (IP-MS) data to identify proteins enriched at least 1.5-fold in the KLHL17-FLAG IP samples as compared to those with the empty vector. This identified 62 proteins as candidate KLHL17 substrates. As KLHL17 is associated with the CRL3 ubiquitin complex, we would expect that KLHL17 substrate expression would decrease with increasing KLHL17 protein levels. Therefore, we next used our global proteomics experiments where we titrated KLHL17 expression in our PANC-1 inducible overexpression system and examined which of the 62 IP enriched proteins decreased with increasing KLHL17 levels. This identified 23 possible substrates including the intermediate filament

proteins vimentin and nestin, both of which are implicated in acinar to ductal metaplasia (ADM), epithelial to mesenchymal transition (EMT) and are induced under inflammatory conditions. Vimentin plays an important role in the recruitment of inflammatory cells to resident tissues, activation of the inflammasome, and fibrosis (reviewed in PMID: 35487686). Nestin is a binding partner of vimentin and a progenitor marker. Nestin⁺ cells have the ability to generate new cells in response to injury (reviewed in PMID: 32393094). We have updated the text under the “Characterizing the function of KLHL17 in the pancreas” section, updated Fig. 5c to show the enrichment of NES and VIM in the IP-MS data, added 4 new panels to Supplementary Fig. 4 (e-h), and added a new supplementary table (Supplementary Table 2) to reflect these findings. We also added a subsection in the Methods outlining our analysis. The updated text (results section, page 5) is highlighted below:

“To identify candidate substrates for degradation by KLHL17, we applied a 1.5-fold enrichment filter (KLHL17-FLAG/Empty Vector) to the KLHL17 IP-MS data and identified 62 proteins (Supplementary Table 2). To further narrow this set of proteins down, we performed a global proteomics experiment where we titrated KLHL17 expression levels in our PANC-1 inducible overexpression system to examine the candidate protein(s) expression as KLHL17 expression increase. As KLHL17 is associated with the CRL3 ubiquitin complex, we would expect that KLHL17 substrates would decrease with increasing KLHL17 protein levels. This narrowed the list of candidate substrates to 23 proteins. Of interest are vimentin/VIM (P -value=0.0029) and nestin (P -value= 0.027) (Fig. 5c, Supplementary Fig. 4e.f., Supplementary Table 2) as both have roles in early pancreatic carcinogenesis and are upregulated in PDAC.²⁸⁻³² In a global proteomics experiments using the MIA PaCa-2 KLHL17 overexpression system, we observed a downward trend in VIM with increasing KLHL17 expression, albeit not significant (P -value=0.17) (Supplementary Fig. 4g, Supplementary Table 2). Nestin, on the other hand, did not display a decrease in expression with increasing KLHL17 (Supplementary Fig. 4h, Supplementary Table 2) in MIA PaCa-2 cells, possibly due to the fact that nestin mRNA²⁷ and protein expression (Supplementary Fig. 4h) is 20-70 fold lower in MIA PaCa-2 than PANC-1. This suggests that KLHL17 may recruit vimentin and nestin to the CRL3 ubiquitin ligase complex for ubiquitination and degradation.”

The authors addressed the limitations of the pancreatic cells lines. The authors should also highlight some of the limitations of the *in silico* KLHL17 knockdown results and their interpretation.

Thank you for bringing this up; we do acknowledge limitations to the *in silico* knockdown method. In the case of the KLHL17 *in silico* knockdown in bulk pancreatic tissue samples from GTEx (n=82 samples in the highest and lowest quartiles) limitations include: 1) a relatively small “knock-down” level as the difference in KLHL17 mRNA levels between the upper and lower quartiles is only 2-fold, and, 2) heterogeneity due to the sample set originating from whole pancreas tissue samples across hundreds of donors. Thus, different factors (e.g. genetic diversity, cell type composition, and environmental factors) may influence the results. We’ve updated the discussion to highlight these limitations (page 7) as follows:

“To uncover *KLHL17*’s role as a functional gene underlying the chr1p36.33 risk signal, we utilized an agnostic approach to start to characterize its function in the pancreas. *In silico* differential gene expression analysis using GTEx pancreas samples comparing those with low KLHL17 to high KLHL17 mRNA expression highlighted an enrichment of upregulated genes involved in inflammation-related pathways and gene sets. This suggests that lower levels of KLHL17 may associate with a pro-inflammatory state in the pancreas, or reduced ability to resolve the consequences of inflammatory signals. While such an *in silico* approach has been shown to have a strong concordance with knockout mouse RNA-sequencing data,³¹ there are limitations to consider when assessing the results. First, for *KLHL17*, there is a relatively small difference in expression between samples in the upper and lower quartile (~2 fold). Second, because the dataset is derived from 168 bulk pancreas tissue samples, heterogeneity (e.g. cell type composition, genetics, environmental factors) is likely to add noise. However, this approach provided us a basis to develop hypotheses regarding KLHL17’s function in the pancreas.”

Reviewer #2 (Remarks to the Author):

In this manuscript, Connelly and colleagues conduct functional dissection of one GWAS risk locus for Pancreatic Cancer. From the tag SNP they first identify additional credible causal SNPs through fine mapping and a Bayesian approach (SuSiE). This is followed by analysis of allele-specific activity using EMSAs, luciferase reporter assays and eQTL analysis. This analysis narrowed it down to three variants and additional analysis of transcription factor binding indicated that rs13303160 represents a functional variant at the locus. They further identify KLHL17 as the likely target gene. This work is timely and is likely to be of interest to a wide community of investigators in functional genomics in cancer.

The introduction is very brief and somewhat disappointing as it fails to put the work in proper context. For example, the reader is not informed whether this is the first GWAS risk locus of pancreatic cancer that has been functionally dissected or there were many other analyzed before. Are there common SNPs between GWAS of PDAC risk and GWAS of traits associated with PDAC risk (obesity, pancreatitis, etc)? In loci identified in previous meta-analyses, was there anything particularly interesting found? Enrichment of pathways?

Thank you for this comment. We have now expanded the background section based on comments from both reviewers. Several PDAC GWAS risk loci indeed share common SNPs or co-localize with GWAS of traits associated with PDAC risk based on the NHGRI-EBI GWAS Catalog and the UKBB/MVP/FinnGen GWAS database. These include: 1q32.1/*NR5A2/T2D*, 2p13.3/*ETAA1/T2D/waist-hip-ratio* (obesity measure), 8q24.21/*MIR1208/T2D*, 9q34/*ABO/T2D/Body fat %*, 12q24.31/*HNF1A/Maturity-onset of diabetes in the young/T2D*, 16q23.1/*BCAR1/T2D*, and 18q21.32/*GRP/T2D/BMI/obesity*. Additionally in the 2018 PDAC GWAS meta-analysis manuscript, pathway enrichment analysis of genes (+/- 100kb) from GWAS loci was performed using different tools. An enrichment of genes associated with Maturity-onset diabetes of the young (KEGG, P -value = 5.5×10^{-9}), Sequence-specific DNA-binding transcription factor activity (GO Molecular Function, P -value = 3.1×10^{-4}), Cellular response to UV (GO Biological Process, P -value = 4.2×10^{-4}) was observed. Further, enrichment analysis using DEPICT to examine which tissues highly express genes associated with GWAS risk loci found numerous gastrointestinal tissues/cell types.

This is now reflected in the introduction (page 2) as follows:

“GWAS have been instrumental in estimating disease risk, identifying candidate genes, and uncovering novel pathways underlying disease development. However, functional characterization of GWAS loci is critical to pinpoint the biological mechanism underlying these signals. Most GWAS signals map to non-coding, regulatory regions of the genome and are hypothesized to influence disease risk through allele specific changes in gene expression.¹⁰ Further, each locus is decorated with tens to hundreds of highly correlated variants. This makes it challenging to identify the functional variant(s) and gene(s) underlying the risk signal. Genomic assays, such as ATAC-seq, histone ChIP-seq, chromatin capture, and massively parallel reporter assays, to map gene regulatory elements and identify putative functional genes and variants have helped improve the process of functionally characterizing GWAS risk loci, however, it is still a time-consuming process. In fact, the biological mechanisms underlying only a handful of the 22 published PDAC risk signals have been functionally characterized to date: 5p15.33/*TERT*,¹¹ 16q23.1/*CTRB2*,¹² and 13q22.1/*DIS3*.¹³

Interestingly, some overlap has been observed between GWAS loci for PDAC and associated epidemiological risk factors. A number of PDAC risk loci have common SNPs or colocalize with GWAS for traits that influence PDAC risk: 1q32.1/*NR5A2/T2D*, 2p13.3/*ETAA1/T2D/waist-hip-ratio* (obesity measure), 8q24.21/*MIR1208/T2D*, 9q34/*ABO/T2D/Body fat percentage*, 12q24.31/*HNF1A/Maturity-onset Diabetes of the Young/T2D*, 16q23.1/*BCAR1/T2D*, 18q21.32/*GRP/T2D/BMI/obesity*.¹⁴ Further pathway enrichment analysis of genes +/- 100kb of PDAC GWAS risk loci indicate an enrichment of genes associated with Maturity-onset diabetes of the young (KEGG, P -value = 5.5×10^{-9}), Sequence-specific DNA-binding transcription factor activity (GO Molecular Function, P -value = 3.1×10^{-4}), Cellular response to UV (GO Biological Process, P -value = 4.2×10^{-4}).⁴ Further, DEPICT enrichment analysis indicated that genes associated with GWAS risk loci are highly expressed in numerous gastrointestinal tissues.⁴⁹

The work is of high quality and the authors are transparent about the technical difficulties to undertake

this type of analysis, especially for such a gene-dense and complex locus. The experiments are conducted with rigor and the interpretation is balanced. The potential chain of causality is well laid out by the authors using assays that are compelling. However, although they identify the likely target gene and propose a plausible scenario for the role of KLHL17 in pancreatic cancer, it is largely speculative. In the absence of additional data tying the role of KLHL17 in inflammation, this excellent paper falls short of being a truly significant advance.

Thank you for your positive comments on our manuscript. We found it important to highlight the limitations and challenges of functional GWAS characterization and the model systems generally used to understand the molecular mechanisms underlying genetic risk loci for pancreatic cancer. We have added results from our experiments attempting to identify potential KLHL17 substrates using a proteomics-based approach. While our results are not validated experimentally, they suggest that KLHL17 may play a role in the degradation of the intermediate filament proteins vimentin and nestin. Both vimentin and nestin are induced upon cell injury and stress; vimentin plays an important role in the recruitment of inflammatory cells to resident tissues, activation of the inflammasome, and fibrosis (reviewed in PMID: 35487686). Nestin is a binding partner of vimentin and a progenitor marker. Nestin⁺ cells have the ability to generate new cells in response to injury (reviewed in PMID: 32393094). Further, both play a role in acinar to ductal metaplasia (ADM) and early epithelial-mesenchymal transition (EMT) prior to pancreatic malignancy. As lower KLHL17 expression are associated with an increased risk of PDAC as per our work, this suggests that vimentin and nestin levels may be elevated (both of which are upregulated in TCGA PAAD) and the injury/inflammatory state maintained rather than resolved. To truly assess an inflammation related phenotype, particularly in ADM and early malignancy, would require an *in vivo* model where multiple cell types can interact in promoting PDAC. We plan to use mouse models to better investigate the effects of heterozygous or homozygous KD of KLHL17, but believe this work is beyond the scope of the current manuscript. We have updated the Results section to include our work and nomination of vimentin and nestin as possible KLHL17 targets. This includes the text below (results section, page 5), along with an updated Fig. 5c, 4 additional panels in Supplementary Table 4 (e-h), and an additional Supplementary table (Supplementary Table 2).

"To identify candidate substrates for degradation by KLHL17, we applied a 1.5-fold enrichment filter (KLHL17-FLAG/Empty Vector) to the KLHL17 IP-MS data and identified 62 proteins (Supplementary Table 2). To further narrow this set of proteins down, we performed a global proteomics experiment where we titrated KLHL17 expression levels in our PANC-1 inducible overexpression system to examine the candidate protein(s) expression as KLHL17 expression increased. As KLHL17 is associated with the CRL3 ubiquitin complex, we would expect that KLHL17 substrates would decrease with increasing KLHL17 protein levels. This narrowed the list of candidate substrates to 23 proteins. Of interest are vimentin/VIM (*P*-value=0.0029) and nestin (*P*-value= 0.027) (Fig. 5c, Supplementary Fig. 4e,f., Supplementary Table 2) as they have both have roles in early pancreatic carcinogenesis and are upregulated in PDAC.²⁸⁻³² In a follow-up global proteomics experiments using the MIA PaCa-2 KLHL17 overexpression system, we do not see a significant decrease in VIM (*P*-value=0.17) between low and high KLHL17 expression, however a downward trend is observed with increasing KLHL17 expression (Supplementary Fig. 4g, Supplementary Table 2). Nestin, on the other hand, does not display a decrease in expression with increasing KLHL17 (Supplementary Fig. 4h, Supplementary Table 2) in MIA PaCa-2 cells, possibly due to the fact that nestin mRNA²⁷ and protein expression (Supplementary Fig. 4h) is 20-70 fold lower as compared to PANC-1 cells. This suggests that KLHL17 may recruit vimentin and nestin to the CRL3 ubiquitin ligase complex for ubiquitination and degradation."

We also expanded our discussion (text below, page 8) to elaborate on the potential role of vimentin and nestin in PDAC risk and the connection to cell injury and inflammation for our hypothesis on how the 1p36.33/*KLHL17* locus may influence PDAC risk. Our model in Fig. 6d has been updated to account for the data from our new analysis.

"Inflammation is a contributing risk factor to the development of PDAC.⁴⁸ PDAC arises from pancreatic intraepithelial neoplasia (PanIN) that display cancerous and pancreatic duct cell properties. Although PanIN exhibit duct-like properties, multiple lines of evidence indicate that pancreatic acinar cells that have undergone acinar-to-ductal metaplasia (ADM) are precursors for PDAC.⁴⁹ ADM is a trans-differentiation

process in which acinar cells lose acinar specific markers and gain duct cell markers. The plasticity of acinar cells makes them highly sensitive to external stimuli.⁵⁰ Acinar cells can recover from an acute stimulus, but with a more sustained stimulus, such as during chronic inflammation, ADM can become irreversible resulting in progression to PanIN.⁵¹ In addition to ADM, epithelial to mesenchymal transition (EMT) is observed in pre-malignant PanIN lesions and is prevalent in regions of ADM with inflammation.^{28,52}

Vimentin and nestin expression, identified in our proteomics as candidate KLHL17 substrates, are induced upon cell injury and stress. Vimentin plays an important role in EMT, the recruitment of inflammatory cells to resident tissues, the activation of the inflammasome, and fibrosis.⁵³ Nestin is a binding partner of vimentin and a marker of multi-potent progenitors. Upon stress and injury, nestin expression is induced and nestin-positive cells have the ability to re-enter the cell cycle and differentiate in the repair process.^{30,31,44,54} In the pancreas, both proteins have been implicated in ADM and pre-malignant EMT and are upregulated in tumors.^{28,30,32} During ADM following pancreatic injury, a transition population of nestin-positive cells is formed.³¹ Additionally, changes in nestin expression correlate with changes in EMT markers.^{29,46} Finally, induction of pancreatitis promotes EMT and vimentin expression.²⁸ As lower expression of KLHL17 is associated with increased risk, this suggests that higher vimentin and nestin expression are associated with an increased risk of PDAC, likely promoting inflammation, ADM and EMT.”

We have not included a response to Reviewer #1's comment but have amended the text to correct the error pointed out by Reviewer #1. We again would like to thank the two reviewers for their helpful suggestions.